# KSR1- and ERK-dependent translational regulation of the epithelial-to-mesenchymal transition

**Chaitra Rao[1], Danielle E Frodyma[1], Siddesh Southekal[2], Robert A Svoboda[3], Adrian R Black[1], Chittibabu Guda[2], Tomohiro Mizutani[4], Hans Clevers[4], Keith R Johnson[1,5], Kurt W Fisher[3], Robert E Lewis[1]\***

[1]Eppley Institute, University of Nebraska Medical Center, Omaha, United States; [2]Department of Genetics, Cell Biology and Anatomy, University of Nebraska Medical Center, Omaha, United States; [3]Department of Pathology and Microbiology, University of Nebraska Medical Center, Omaha, United States; [4]Hubrecht Institute, Royal Netherlands Academy of Arts and Sciences (KNAW) and UMC Utrecht, Utrecht, Netherlands; [5]Department of Oral Biology, University of Nebraska Medical Center, Omaha, United States

**\*For correspondence:** rlewis@unmc.edu

**Competing interest:** The authors declare that no competing interests exist.

**Abstract:** The epithelial-to-mesenchymal transition (EMT) is considered a transcriptional process that induces a switch in cells from a polarized state to a migratory phenotype. Here, we show that KSR1 and ERK promote EMT-like phenotype through the preferential translation of Epithelial-Stromal Interaction 1 (EPSTI1), which is required to induce the switch from E- to N-cadherin and coordinate migratory and invasive behavior. EPSTI1 is overexpressed in human colorectal cancer (CRC) cells. Disruption of KSR1 or EPSTI1 significantly impairs cell migration and invasion in vitro, and reverses EMT-like phenotype, in part, by decreasing the expression of N-cadherin and the transcriptional repressors of E-cadherin expression, ZEB1 and Slug. In CRC cells lacking KSR1, ectopic EPSTI1 expression restored the E- to N-cadherin switch, migration, invasion, and anchorage-independent growth. KSR1-dependent induction of EMT-like phenotype via selective translation of mRNAs reveals its underappreciated role in remodeling the translational landscape of CRC cells to promote their migratory and invasive behavior.

## Introduction

Molecular scaffolds affect the intensity and duration of signaling pathways by coordinating a discrete set of effectors at defined subcellular locations to regulate multiple cell fates (*Morrison and Davis, 2003*; *Pawson and Scott, 1997*). Kinase Suppressor of Ras 1 (KSR1) serves as a scaffold for Raf, MEK, and ERK enabling the efficient transmission of signals within the mitogen activated protein kinase (MAPK) cascade (*Kortum and Lewis, 2004*; *Nguyen et al., 2002*). Although KSR1 is dispensable for normal development, it is necessary for oncogenic Ras-induced tumorigenesis including colorectal cancer cells (*Kortum and Lewis, 2004*; *Nguyen et al., 2002*; *Fisher et al., 2011*; *Fisher et al., 2015*; *Morrison et al., 2009*; *Rao et al., 2020*), suggesting that KSR1 may modulate aberrant signals that redirect the function of effectors typically involved in normal cellular homeostasis. Activating Ras mutations are present in over 40 % of colorectal cancers (CRC), and associated with advanced disease and decreased overall survival (*Haigis, 2017*; *Serebriiskii et al., 2019*). Activated Ras, a critical driver of both tumor growth and survival, is an alluring therapeutic target, yet targeting the majority of oncogenic Ras alleles is still a work in progress. Raf/MEK/ERK signaling can phenocopy Ras signaling essential for CRC growth and survival (*Schmitz et al., 2007*; *Brandt et al., 2019*). Therefore, understanding

**eLife digest** The majority of cancer deaths result from tumor cells spreading to other parts of the body via a process known as metastasis. 90% of all cancers originate in epithelial cells that line the inner and outer surface of organs in our bodies. Epithelial cells, however, are typically stationary and must undergo various chemical and physical changes to transform in to migratory cells that can invade other tissues.

This transformation process alters the amount of protein cells use to interact with one another. For example, epithelial cells from the colon produce less of a protein called E-cadherin as they transition into migrating cancer cells and make another protein called N-cadherin instead. A protein called KSR1 is a key component of a signaling pathway that is responsible for generating the proteins colon cancer cells need to survive. But it is unknown which proteins KSR1 helps synthesize and whether it plays a role in the metastasis of colon cancer cells.

To investigate this, Rao et al. studied the proteins generated by cancerous colon cells cultured in the laboratory, in the presence and absence of KSR1. The experiment showed that KSR1 increases the levels of a protein called EPSTI1, which colon cancer cells need to transform into migratory cells. Depleting KSR1 caused cancer cells to generate less EPSTI1 and to share more features with healthy cells, such as higher levels of E-cadherin on their surface and reduced mobility. Adding EPSTI1 to the cancer cells that lacked KSR1 restored the traits associated with metastasis, such as high levels of N-cadherin, and allowed the cells to move more easily.

These findings suggest that KSR1 and EPSTI1 could be new drug targets for reducing, or potentially reversing, the invasive behavior of colon cancer cells. However, further investigation is needed to reveal how EPSTI1 is generated and how this protein helps colon cancer cells move and invade other tissues.

the effectors that transmit signals emanating from oncogenic Ras is a valuable step in detecting and targeting the pathways critical to tumor cell function and their adaptation to therapy.

Oncogene-driven signaling pathways promote mRNA translation that enables expression of a subset of mRNAs to promote growth, invasion, and metastasis (*Chu et al., 2016*; *Avdulov et al., 2004*; *Truitt Morgan et al., 2015*; *Pelletier et al., 2015*). Tumor cells have an increased dependence on cap-dependent translation, unlike their normal complements (*Truitt Morgan et al., 2015*; *Truitt and Ruggero, 2016*). Eukaryotic Translation Initiation Factor 4E (eIF4E) is a rate-limiting factor for oncogenic transformation, with reductions of as little as 40 % being sufficient to block tumorigenesis (*Truitt Morgan et al., 2015*). eIF4E function is regulated by association of 4E-binding proteins (4EBPs). Importantly, disruption of KSR1 or ERK inhibition leads to dephosphorylation and activation of 4EBP1, indicating that the function of KSR1 as an ERK scaffold is key to the aberrant regulation of mRNA translation (*McCall et al., 2016*). This tumor-specific, KSR1-dependent regulation of mRNA translation of a subset of genes was predicted to selectively promote survival of CRC cells but not normal colon epithelia (*McCall et al., 2016*; *Neilsen et al., 2019*).

Almost all CRC originates from epithelial cells lining the colon or rectum of the gastrointestinal tract, but in order to invade to the surrounding tissue, cancer cells lose cell adhesiveness to acquire motility and become invasive, characterized by the epithelial-to-mesenchymal transition (EMT), which is central to tumor pathogenesis (*Ye and Weinberg, 2015*; *Nieto, 2013*; *Thiery et al., 2009*; *Dongre and Weinberg, 2019*). EMT involves a complex cellular process during which epithelial cells lose polarity, cell-cell contacts and acquire mesenchymal characteristics. While EMT is crucial for cell plasticity during embryonic development, trans differentiation and wound healing, when aberrantly activated EMT has deleterious effects, which facilitate motility and invasion of cancer cells (*Nieto, 2013*; *Thiery et al., 2009*; *Dongre and Weinberg, 2019*; *Nieto et al., 2016*). EMT has been shown to be controlled by transcription-dependent mechanisms, especially through repression of genes that are hallmarks of epithelial phenotype such as E-cadherin. Loss of E-cadherin at the membrane has been associated with carcinoma progression and EMT (*Thiery et al., 2009*; *Thiery, 2002*; *Oda et al., 1994*; *Frixen et al., 1991*). E-cadherin function is transcriptionally repressed through the action of EMT transcription factors (TFs), including Snail-family proteins (*Snail1*, *Slug*), zinc finger E-box binding homeobox 1 and 2 (*ZEB1* and *ZEB2*), and twist-related protein (*Twist*) (*Nieto et al., 2016*; *Jolly et al.,*

*2017*). Transcriptional control of E-cadherin is unlikely to be sole determinant of EMT, invasion and metastasis. Inappropriate induction of non-epithelial cadherins, such as N-cadherin by epithelial cells are known to play a fundamental role during initiation of metastasis (*Nieman et al., 1999*; *Liu et al., 2017*; *Suyama et al., 2002*; *Rosivatz et al., 2004*; *Okubo et al., 2017*; *Sadot et al., 1998*; *Loh et al., 2019*). N-cadherin disassembles adherent junction complexes, disrupting the intercellular cohesion and reorienting the migration of cells, away from the direction of cell-cell contact (*Nieman et al., 1999*; *Scarpa et al., 2015*). Upregulation of N-cadherin expression promotes motility and invasion (*Nieman et al., 1999*; *Liu et al., 2017*; *Suyama et al., 2002*; *Hulit et al., 2007*). Thus, central to the process of EMT is the coordinated loss of E-cadherin expression and the upregulation of N-cadherin gene expression, termed cadherin switching (*Loh et al., 2019*; *Wheelock et al., 2008*; *Tomita et al., 2000*; *Maeda et al., 2005*; *Araki et al., 2011*).

Previous studies have demonstrated transcriptional regulation of EMT through oncogenic Ras or its downstream effector signaling pathways via the activation of EMT-TFs (*Shin et al., 2010*; *Shin et al., 2019*; *Andreolas et al., 2008*; *Liu et al., 2014*; *Wong et al., 2013*; *Wang et al., 2010*; *Lemieux et al., 2009*). Oncogenic Ras itself activates EMT-TF *Slug* to induce EMT in skin and colon cancer cells (*Wong et al., 2013*; *Wang et al., 2010*). Enhanced activity of ERK2 but not ERK1, has been linked with Ras-dependent regulation of EMT (*Shin et al., 2010*; *Shin et al., 2019*). Several studies have also described an alternative program wherein cells lose their epithelial phenotype, via post-transcriptional modifications rather than transcriptional repression involving translational regulation or protein internalization (*Jechlinger et al., 2003*; *Aiello et al., 2018*; *Waerner et al., 2006*). Expression profiling of polysome-bound mRNA to assess translational efficiency identified over 30 genes that were translationally regulated upon Ras and *TGFβ* inducing EMT (*Jechlinger et al., 2003*; *Waerner et al., 2006*). Functional characterization of the resultant proteins should reveal preferentially translated mRNAs essential to invasion and metastasis.

*EPSTI1* was identified as a stromal fibroblast-induced gene upon co-cultures of breast cancer cells with stomal fibroblasts (*Nielsen et al., 2002*). EPSTI1 is expressed at low levels in normal breast and colon tissue but aberrantly expressed in breast tumor tissue (*Nielsen et al., 2002*). EPSTI1 promotes cell invasion and malignant growth of primary breast tumor cells (*Li et al., 2014*; *de Neergaard et al., 2010*). We performed polysome profiling in CRC cells and found that KSR1- and ERK induces of EPSTI1 mRNA translation. EPSTI1 is both necessary and sufficient for coordinating the upregulation of N-cadherin with the downregulation of E-cadherin to stimulate cell motility and invasion in colon cancer cells. These data demonstrate that ERK-regulated regulation of mRNA translation is an essential contributor to EMT-like phenotype and reveal a novel effector of the cadherin switch whose characterization should yield novel insights into the mechanisms controlling the migratory and invasive behavior of cells.

## Results

### Genome-wide polysome profiling reveals translational regulation of EPSTI1 by KSR1

ERK signaling regulates global and selective mRNA translation through RSK1/2-dependent modification of cap-dependent translation (*McCall et al., 2016*; *Roux et al., 2007*). Phosphorylation of cap binding protein 4E-BP1 releases eIF4E to promote translation and the abundance of eIF4E is a rate-limiting factor for oncogenic Ras- and Myc-driven transformation (*Truitt Morgan et al., 2015*). We showed previously that KSR1 maximizes ERK activation in the setting of oncogenic Ras (*Kortum et al., 2006*), which is required for increased Myc translation via dephosphorylation of 4E-BP1, supporting CRC cell growth (*McCall et al., 2016*). These observations imply that the ERK scaffold function of KSR1 alters the translational landscape in CRC cells to support their survival.

To determine the effect of KSR1 on translatomes in colon cancer cells, we performed genome-wide polysome profiling (*King and Gerber, 2016*). We stably expressed short hairpin RNA (shRNA) constructs targeting KSR1 (KSR1 RNAi) or a non-targeting control in two K-Ras mutant CRC cell lines, HCT116 and HCT15 (*Figure 1D*, top panels). We isolated and quantified both total mRNA and efficiently translated mRNAs (associated with ≥3 ribosomes) using RNA sequencing (*Figure 1A*, *Figure 1—figure supplement 1*). We used Anota2seq (*Oertlin et al., 2019*) to calculate translation efficiency (TE) by comparing the differences in efficiently translated mRNAs to the total transcript of

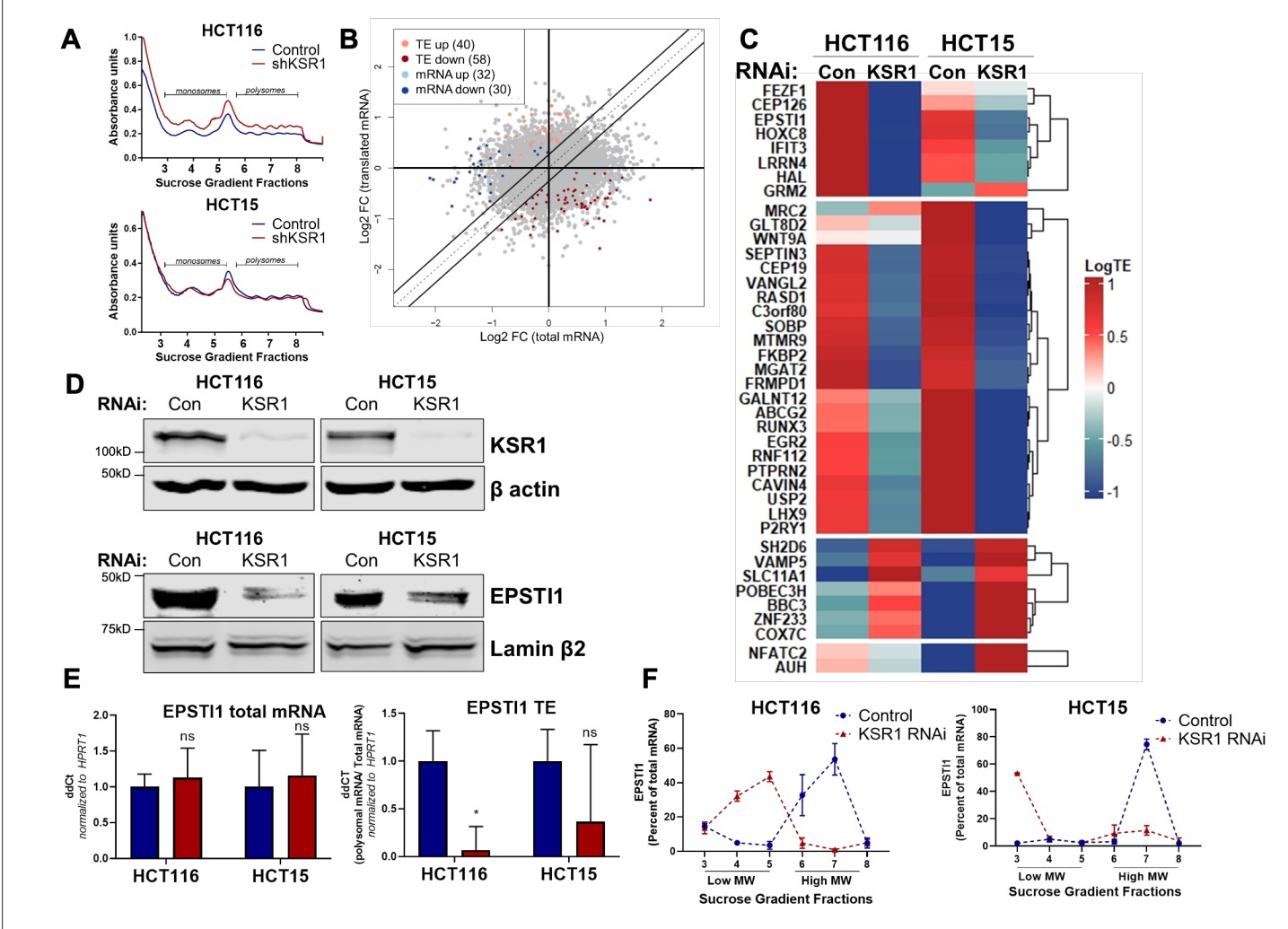

**Figure 1.** EPSTI1 translation is regulated by KSR1. (**A**) Representative polysome profiles from control and KSR1 knockdown (KSR1 RNAi) HCT116 and HCT15 cells. Sucrose gradient fractions 3–5 denote the low-molecular-weight complexes (monosomes) and the fractions 6–8 are the high-molecular-weight complexes (polysomes). (**B**) Scatter plot of polysome-associated mRNA to total mRNA log2 fold-changes upon KSR1 knockdown in HCT116 and HCT15 with RNA-seq. The statistically significant genes in the absence of KSR1 are classified into four groups with a fold change ($|\log_2\text{FC}|$) > 1.2 and p-value < 0.05. The number of mRNAs with a change in TE (orange and red) are indicated (n = 3 for each condition). TE, translational efficiency. (**C**) Heatmap of TE changes for the top 40 RNAs control and KSR1 knockdown (KSR1 RNAi) HCT116 and HCT15 cells (n = 3 for each condition). (**D**) Western blot analysis of KSR1 and EPSTI1 following KSR1 knockdown in HCT116 and HCT15 cells. (**E**) RT-qPCR analysis of EPSTI1 mRNA from total RNA and polysomal RNA (fractions number 6–8) in control and KSR1 knockdown HCT116 and HCT15 cells, the TE was calculated as the ratio of polysomal mRNA to the total mRNA (n = 3; *, p < 0.05). (**F**) RT-qPCR analysis of *EPSTI1* mRNA levels isolated from sucrose gradient fractions of the control and KSR1 knockdown HCT116 and HCT15 cells. Fractions 3–5 (low MW) and 6–8 (high MW) are plotted for the control and KSR1 knockdown state with values corresponding to the percentage of total mRNA across these fractions n = 3. Experiments shown in (**A–F**) are representative of three independent experiments.

The online version of this article includes the following figure supplement(s) for figure 1:

**Figure supplement 1.** Three independent replicates of the polysome profiles from control and KSR1 knockdown (shKSR1) HCT116 and HCT15 cells.

**Figure supplement 2.** EPSTI1 is translationally regulated by KSR1.

each mRNA and observed that a significant number of mRNAs ([selDeltaTP ≥ log (1.2) and selDeltaPT ≥ log (1.2)] and p-value < 0.05) showed either reduced TE or upregulated TE upon KSR1 disruption (*Figure 1B, C*, *Supplementary file 1*, *Source data 1*) in both HCT116 and HCT15 cells. Gene Set Enrichment Analysis (GSEA) (*Subramanian et al., 2005*) of significantly enriched genes in HCT116 and HCT15 (*Figure 1B*, *Figure 1—figure supplement 2A, B*), identified 11 mRNAs in the gene set titled "Hallmark EMT signature", "Jechlinger EMT Up", and "Gotzmann EMT up" , that had significantly

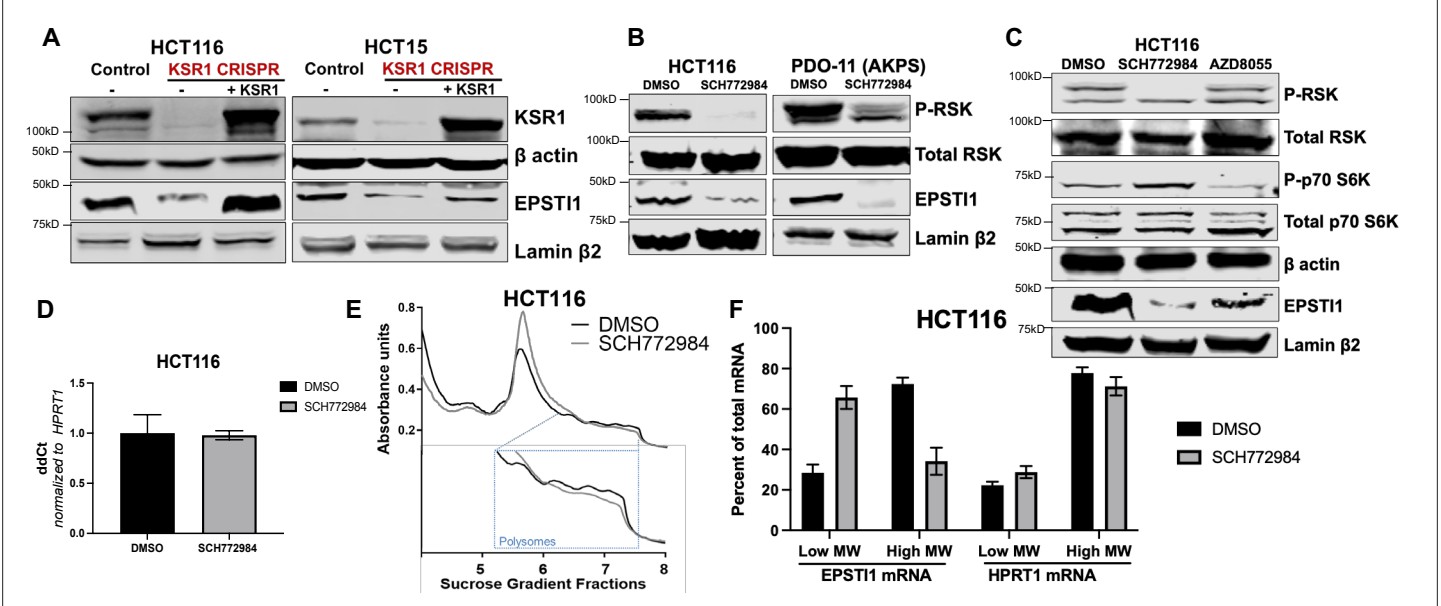

**Figure 2.** KSR1 or ERK inhibition suppresses EPSTI1 protein expression in cell lines and organoids. (**A**) Cell lysates prepared from control, KSR1 CRISPR-targeted (KSR1 CRISPR) and CRISPR-targeted HCT116 and HCT15 cells expressing KSR1 (KSR1 CRISPR+ KSR1) analyzed for EPSTI1 protein expression by western blotting. (**B**) Western blot of the indicated proteins in HCT116 (left) and AKPS quadruple mutant organoids (right) treated with DMSO or 1 µM of SCH772984 for 48 hr. (**C**) EPSTI1 protein expression was analyzed by western blot in HCT116 cells treated with DMSO, 1 µM of SCH772984, or 1 µM of AZD8055 for 48 hr. (**D**) RT-qPCR analysis of *EPSTI1* mRNA from total RNA in HCT116 cells treated with either DMSO or ERK1/2 selective inhibitor, SCH772984 (n = 3; ns, non-significant). (**E**) Representative polysome profiles from HCT116 cells treated DMSO or 1 µM of ERK1/2 selective inhibitor, SCH772984. (**F**) RT-qPCR analysis of *EPSTI1* and *HPRT1* mRNA levels from LMW (fractions 3–5) and HMW (fractions 6–8) of the DMSO control or SCH772984-treated HCT116 cells (n = 3; *, p < 0.05; ***, p < 0.001). All values displayed as mean ± S.D. Experiments shown in (**A–F**) are representative of three independent experiments.

decreased translation upon KSR1 disruption (*Supplementary file 2*). Among the genes with decreased translation, *EPSTI1* was one of the highly significant mRNAs. We sought to determine the functional relevance of KSR1-dependent induction of EPSTI1 to phenotypic plasticity in colon cancer cells.

To confirm that EPSTI1 translation is KSR1-dependent, we observed that, EPSTI1 protein expression was decreased with the knockdown of KSR1 in HCT116 and HCT15 cells (*Figure 1D*), while the total mRNA transcript was unchanged upon KSR1 disruption (*Figure 1E*, left panel). *EPSTI1* TE was markedly decreased upon KSR1 depletion (*Figure 1E*, right). RT-qPCR analysis of sucrose-gradient fractions of monosome mRNA and polysome RNA distribution confirmed that *EPSTI1* mRNA shifted from actively translating high-molecular-weight (MW) polysome fractions to low-MW fractions in KSR1 knockdown cells (*Figure 1F*). In contrast, *HPRT1* mRNA was insensitive to KSR1 knockdown in HCT116 and HCT15 cells, and qPCR analysis of *HPRT1 mRNA* isolated from sucrose gradient fractions of control and KSR1 knockdown cells showed no significant shift between the low-MW and the high-MW fractions (*Figure 1—figure supplement 2C*). To determine if KSR1 promotes EPSTI1 degradation, we first assessed EPSTI1 turnover in HCT116 cells following treatment with a protein-synthesis inhibitor, cycloheximide (CHX) and observed that EPSTI1 has a 6 hr half-life (*Figure 1—figure supplement 2D*). We analyzed EPSTI1 turnover using a combination of proteasome inhibitor, MG132 and CHX in control and CRISPR-targeted KSR1 HCT116 cells (*Figure 1—figure supplement 2E*). EPSTI1 turnover was not sensitive to MG132 treatment in HCT116 cells lacking KSR1 expression. Therefore, in HCT116 cells, KSR1 does not mediate ubiquitin proteosome system (UPS)-mediated degradation of EPSTI1. These data support our conclusion that EPSTI1 translation is induced by KSR1.

## KSR1/ERK signaling regulates EPSTI1 expression in colon cancer cells

To confirm our observations in KSR1 knockdown cells, we tested the effect of CRISPR/Cas9-mediated targeting of KSR1 on EPSTI1 in CRC cell lines. EPSTI1 protein expression was decreased upon KSR1 depletion in HCT116 and HCT15 cells and EPSTI1 expression was restored in knockout cells upon expression of a KSR1 transgene (+ KSR1) (*Figure 2A*). Similar to inhibition of KSR1, treatment with

ERK inhibitor SCH772984 (*Morris et al., 2013*) suppressed EPSTI1 protein expression in both CRC cell line HCT116 and tumorigenic patient derived colon organoid engineered with deletion of APC, p53, SMAD4, and K-Ras$^{G12D}$ mutation (PDO-11 AKPS) (*Figure 2B*; *Drost et al., 2015*). To determine if EPSTI1 expression is also dependent on mTOR signaling, we tested the effect of mTOR inhibition on EPSTI1expression. Though mTOR inhibitor, AZD8055 (*Chresta et al., 2010*) robustly inhibited phosphorylation of mTOR substrate p70 S6 kinase, its ability to decrease EPSTI1 expression in HCT116 cells was weak relative to treatment with the ERK inhibitor (*Figure 2C*). These observations suggest the ERK affects EPSTI1 expression via mechanisms distinct from mTOR. While the total protein was reduced upon ERK inhibition in HCT116, the *EPSTI1* transcript levels were not altered significantly by SCH772984 treatment (*Figure 2D*).

We performed polysome profiling in HCT116 cells, either treated with DMSO or ERK inhibitor, SCH772984 and we isolated mRNA from low-MW monosome (fractions 3–5) and high-MW polysome (fractions 6–8) fractions (*Figure 2E*). RT-qPCR demonstrated that *EPSTI1* mRNA shifted from high-MW fractions to the low-MW fractions upon ERK inhibition (*Figure 2F*). The distribution of mRNA for *HPRT1* within the same profile was not altered by SCH772984 treatment (*Figure 2F*). These data

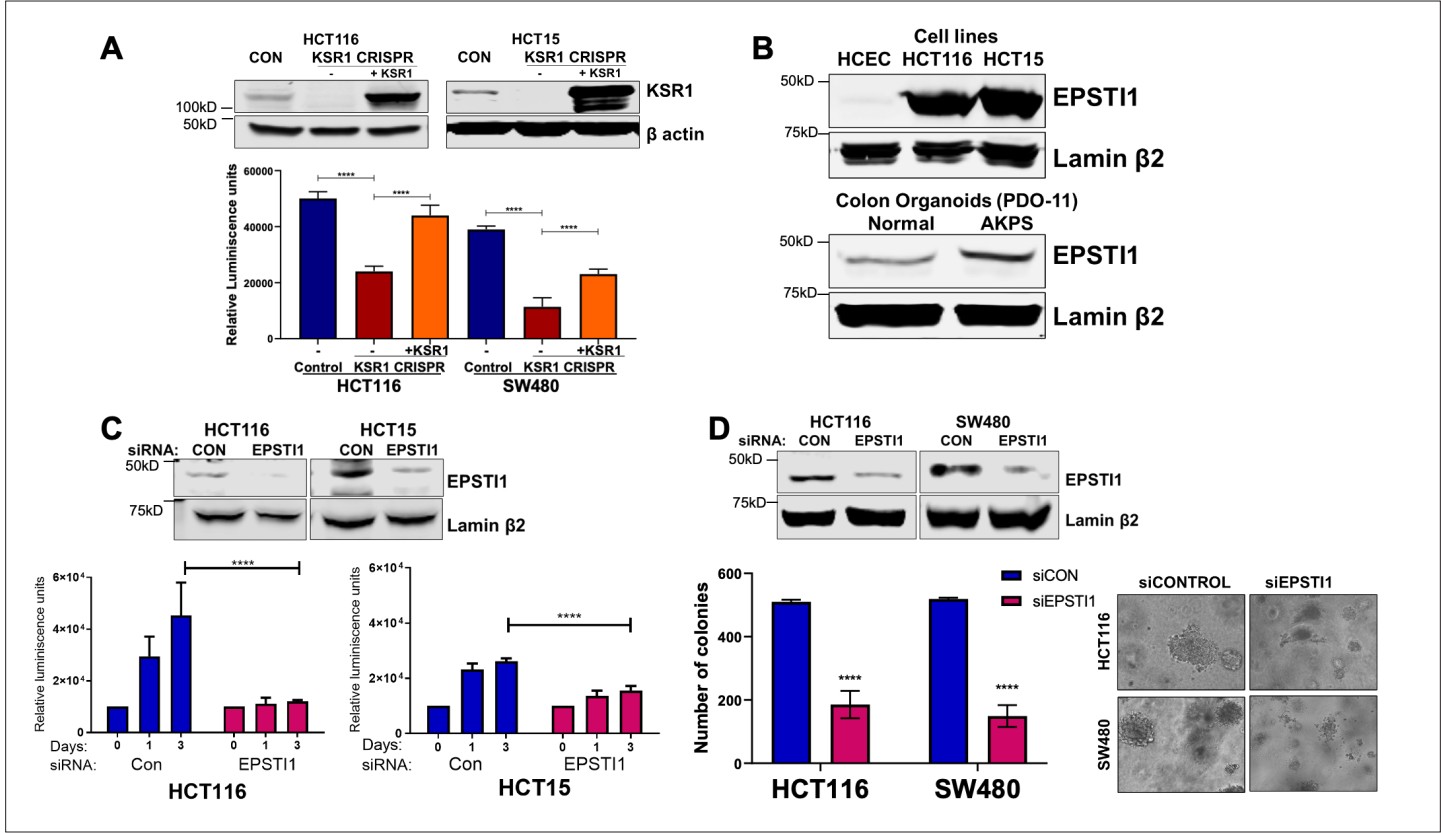

**Figure 3.** EPSTI1 is overexpressed in cancer cell lines and organoids and promotes anchorage-independent growth. (**A**) Anchorage-independent cell viability was analyzed in HCT116 and HCT15 cells plated on poly-(HEMA)-coated plates was measured using CellTiter-Glo following CRISPR-targeting (KSR1 CRISPR) and re-expressing KSR1 (KSR1 CRISPR+ KSR1) in the CRISPR-targeted cells. The data are shown as relative luminescence units mean ± SD, n = 6. Matched results were analyzed for statistical significance one-way ANOVA followed by t-test. (Upper panels) Western blot showing the expression of KSR1 in control, KSR1 knockout and KSR1-knockout cells expressing a KSR1 transgene (+ KSR1). (**B**) Western blot analysis of EPSTI1 protein expression was assessed in HCECs, HCT116, HCT15, normal human colon organoids, and transformed AKPS colon organoids. (**C**) Viability of HCT116 and HCT15 cells measured using CellTiter-Glo following siRNA knockdown of EPSTI1 that were plated on poly-(HEMA)-coated plates to simulate anchorage-independent conditions. Cell viability was measured immediately after plating and 0, 1, and 3 days after plating (n = 6). The data are shown as mean luminescence units ± SD. Matched results were analyzed for statistical significance by t-test. (Top) Western blot confirming the knockdown of EPSTI1 in HCT116 and SW480 at Day 3. (**D**) (Left) Quantification of the colonies formed in HCT116 and SW480 cells following RNAi knockdown using non-targeting control (siCON) or EPSTI1 (siEPSTI1) after plating on soft agar. (Right) Representative photomicrographs of colonies for each sample. The data are illustrated as the number of colonies present after 2 weeks, mean ± SD, n = 6. Paired results were analyzed for statistical significance using Student's *t* test. (Top) Western blot confirming the knockdown of EPSTI1 in HCT116 and SW480 cells. ****, p < 0.0001.

indicate that KSR1-dependent ERK signaling is a critical regulator of EPSTI1 mRNA translation in colon cells and organoids.

## EPSTI1 is required for anchorage-independent growth in colon cancer cells

KSR1 disruption inhibits HCT116 cell anchorage-independent growth in vitro and tumor formation in vivo (*Fisher et al., 2015*). Similarly, disruption of KSR1 by CRISPR/Cas9-mediated targeting decreased HCT116 and HCT15 cell viability under anchorage-independent conditions on simulated by poly-(HEMA) coating (*Figure 3A*). KSR1 transgene expression restored cell viability in HCT116 and HCT15 cells lacking KSR1 (KSR1 CRISPR+ KSR1) (*Figure 3A*). We showed previously that KSR1 expression is upregulated in colon cancer cell lines when compared to the non-transformed human colon epithelial cells (HCECs) (*Fisher et al., 2015*). We observed that EPSTI1 protein is aberrantly expressed in colon cancer cell lines HCT116 and HCT15, while its expression is detected weakly in HCECs (*Figure 3B*). EPSTI1 protein expression is also markedly higher in AKPS organoids than normal colon organoids (*Figure 3B*).

To determine the regulation of EPSTI1 in human colon tumor maintenance, we performed siRNA knockdown of EPSTI1 in HCT116 and HCT15 cells. EPSTI1 disruption suppressed viability on poly-(HEMA) coated by 40 % in HCT15 cells, and over 70%, in HCT116 cells (*Figure 3C*). EPSTI1 knockdown reduced colony formation in soft agar by 63 % in HCT116 cells and 71 % in SW480 cells (*Figure 3D*). These observations show that KSR1-dependent translation of ESPTI1 is required for anchorage-independent growth of colon tumor cell lines.

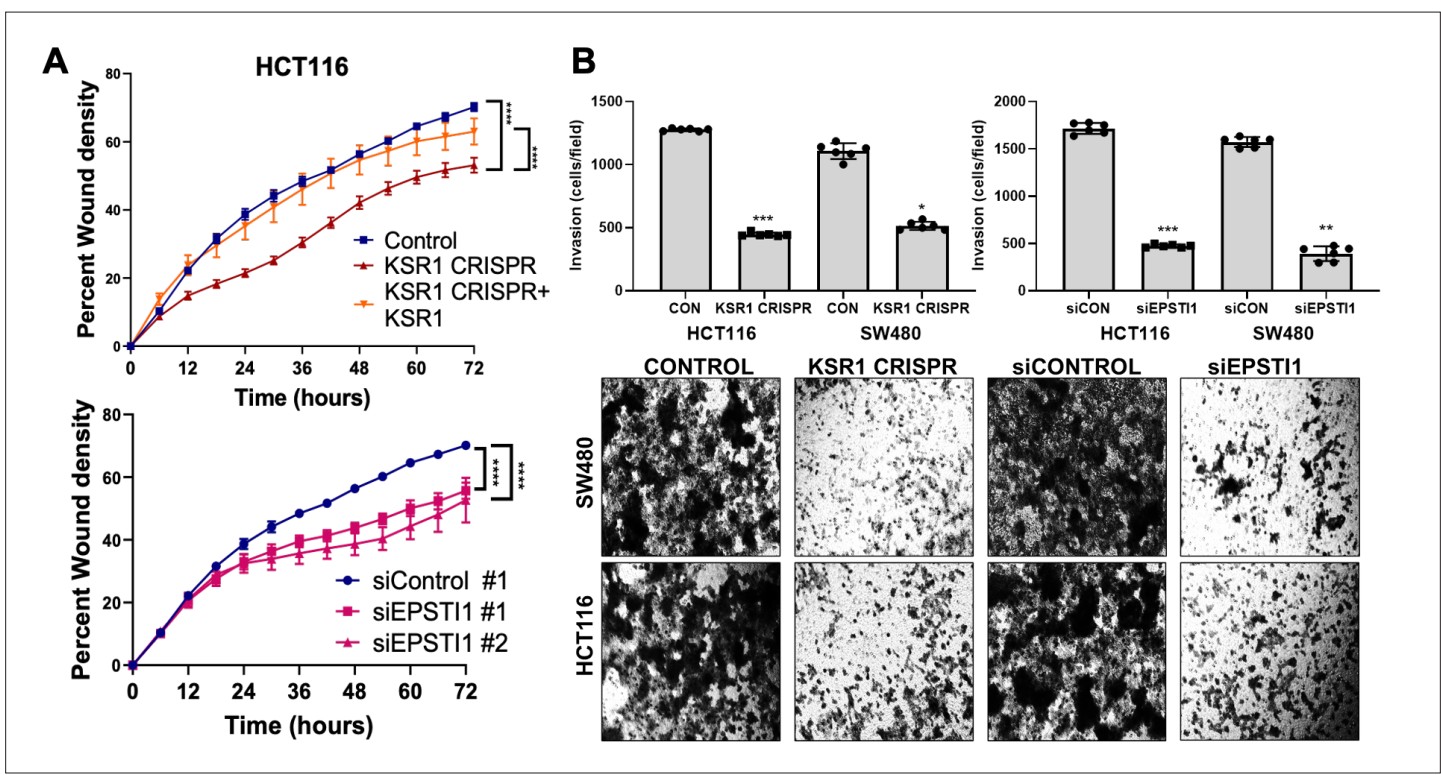

**Figure 4.** KSR1 or EPSTI1 promote migration and invasion in CRC cells. (**A**) Control, CRISPR-targeted (KSR1 CRISPR) and CRISPR-targeted HCT116 cells expressing KSR1 (KSR1 CRISPR+ KSR1) (upper) and control or EPSTI1 knockdown HCT116 cells (lower) were evaluated in a 96-well IncuCyte scratch wound assay. The graph represents the time kinetics of percent wound density, calculated by IncuCyte ZOOM software, shown as mean ± SD, n = 12 ****, p < 0.0001. Matched results were analyzed for statistical significance using one-way ANOVA with Dunnett's posttest for multiple comparisons. (**B**) (Upper panels) Control, KSR1 knockout (KSR1 CRISPR), and EPSTI1 knockdown (siEPSTI1) were subjected to Transwell migration assay through Matrigel for 24 hr using 10 % FBS as chemoattractant. The number of invaded cells per field were counted. Data are the mean ± SD (n = 6); *, p < 0.1; **, p < 0.01; ***, p < 0.001. (Lower panels) Representative images of Giemsa-stained cells 24 hr after invasion through Matrigel.

## KSR1 or EPSTI1 disruption decreases cell mobility in CRC cells

Considering the suggested role of EPSTI1 in promoting EMT-like phenotypes (*Nielsen et al., 2002*; *Li et al., 2014*), we sought to evaluate the biological role of EPSTI1 in colon cancer cells. Time-lapse images of control and EPSTI1 knockdown in HCT116 cell motility in a scratch wound was analyzed by measuring the relative wound density (*Johnston et al., 2015*) over 72 hours (*Figure 4A*, bottom). IncuCyte software was used to calculate relative wound density, that is, the percentage of spatial cell density inside the wound relative to the spatial density outside of the wound area at a given time point. The calculation of cell migration using this method, avoids false changes in cell density due to proliferation. Motility was also assessed in control, CRISPR-targeted (KSR1 CRISPR), and CRIPSR-targeted HCT116 cells expressing KSR1 (KSR1 CRISPR+ KSR1) (*Figure 4A*, top). Cells lacking either EPSTI1 or KSR1 were approximately 20 % less motile compared to control cells. Reintroduction of KSR1 expression in CRISPR-targeted HCT116 cells restored motility comparable to the control cells (*Figure 4A*, top).

EPSTI1 knockdown HCT116 and SW480 cells were subjected to Transwell invasion assays. EPSTI1 RNAi suppresses cell invasion through Matrigel by 72 % in HCT116 and by 75 % in SW480. (*Figure 4B*, top right and bottom). Since KSR1 is required for EPSTI1 translation, we determined the functional contribution of KSR1 in regulating cell invasion. KSR1 depletion suppressed invasion by 64 % in HCT116 and by 53 % SW480 cells (*Figure 4B*, top left and bottom). Overall, these results suggest the KSR1-dependent EPSTI1 signaling contributes to cell migration and invasion in CRC cells.

## KSR1 or EPSTI1 disruption causes cadherin switching in CRC cells

To understand the underlying mechanism by which KSR1 or EPSTI1 promote motility and invasion in CRC cells, we evaluated their contribution to the expression of critical determinants of EMT that modulate cell adhesion, E- and N-cadherins and EMT-TFs. Compared to the non-targeting control, KSR1 disruption in HCT116, HCT15 and SW480 cells had elevated levels of E-cadherin, along with a coincident decrease in EMT-TF Slug (*Figure 5A*). Expression of Vimentin, and Snail1 was not changed in HCT116 cells (*Figure 5—figure supplement 1A*). Upon knockdown of EPSTI1 with either of two siRNA oligos, we observed a decrease in the expression of N-cadherin, ZEB1 and Slug. Coincident with the decrease in EMT-TFs, E-cadherin levels were elevated (*Figure 5B*). While there was no significant change in the Slug and ZEB1 mRNA upon EPSTI1 knockdown (*Figure 5—figure supplement 1B*), EPSTI1 disruption decreased N-cadherin mRNA expression over 50 % in HCT116 and SW480 cells (*Figure 5C*). Following EPSTI1 knockdown, we subjected control HCT116 cells and HCT116 cells overexpressing N-cadherin to Transwell invasion assay through Matrigel. EPSTI1 knockdown suppressed cell invasion. The expression of N-cadherin in cells lacking EPSTI1 was sufficient to restore invasiveness to HCT116 cells (*Figure 5—figure supplement 1C, D*). This is consistent with previous observations that upregulation of N-cadherin expression enhances motility in multiple cancer cell lines (*Nieman et al., 1999*; *Hulit et al., 2007*; *Mrozik et al., 2018*). These results indicate that the switch of E-cadherin to N-cadherin expression promotes the progression of migratory and invasive behavior orchestrated by EPSTI1 signaling in CRC cells.

## EPSTI1 is necessary and sufficient for EMT-like phenotype in CRC cells

To determine the extent to which KSR1- and ERK-dependent EPSTI1 translation is critical to colon tumor cell growth and invasion, we expressed a MSCV-FLAG-EPSTI1-GFP construct in KSR1-CRISPR knockout HCT116, SW480, and HCT15 cells. CRISPR/Cas9-mediated deletion of KSR1 disrupted EPSTI1 expression, downregulated Slug and N-cadherin expression and elevated E-cadherin expression (*Figure 6A*). E-cadherin staining was absent in control CRC cells but evident at the cell membrane in KSR1 knockout cells (*Figure 6B*). Exogenous expression of EPSTI1 in cells lacking KSR1 restored the cadherin switch, by decreasing the expression of E-cadherin (*Figure 6A, B*) and increasing N-cadherin levels comparable to control cells (*Figure 6A*). Suppression of E-cadherin and restoration of N-cadherin expression by the EPSTI1 transgene reestablished the ability of KSR1 knockout cells to migrate in monolayer culture (*Figure 6C*) and invade through Matrigel. Forced expression of EPSTI1 in these cells, increased the number of invading cells by over threefold (*Figure 6D*). To determine the effect of EPSTI1 on cell proliferation, we analyzed the cell growth kinetics in HCT116 and SW480 cells (*Figure 6—figure supplement 1*). Over 3 days, EPSTI1 knockdown had no effect on cell proliferation compared to control HCT116 and SW480 cells. While

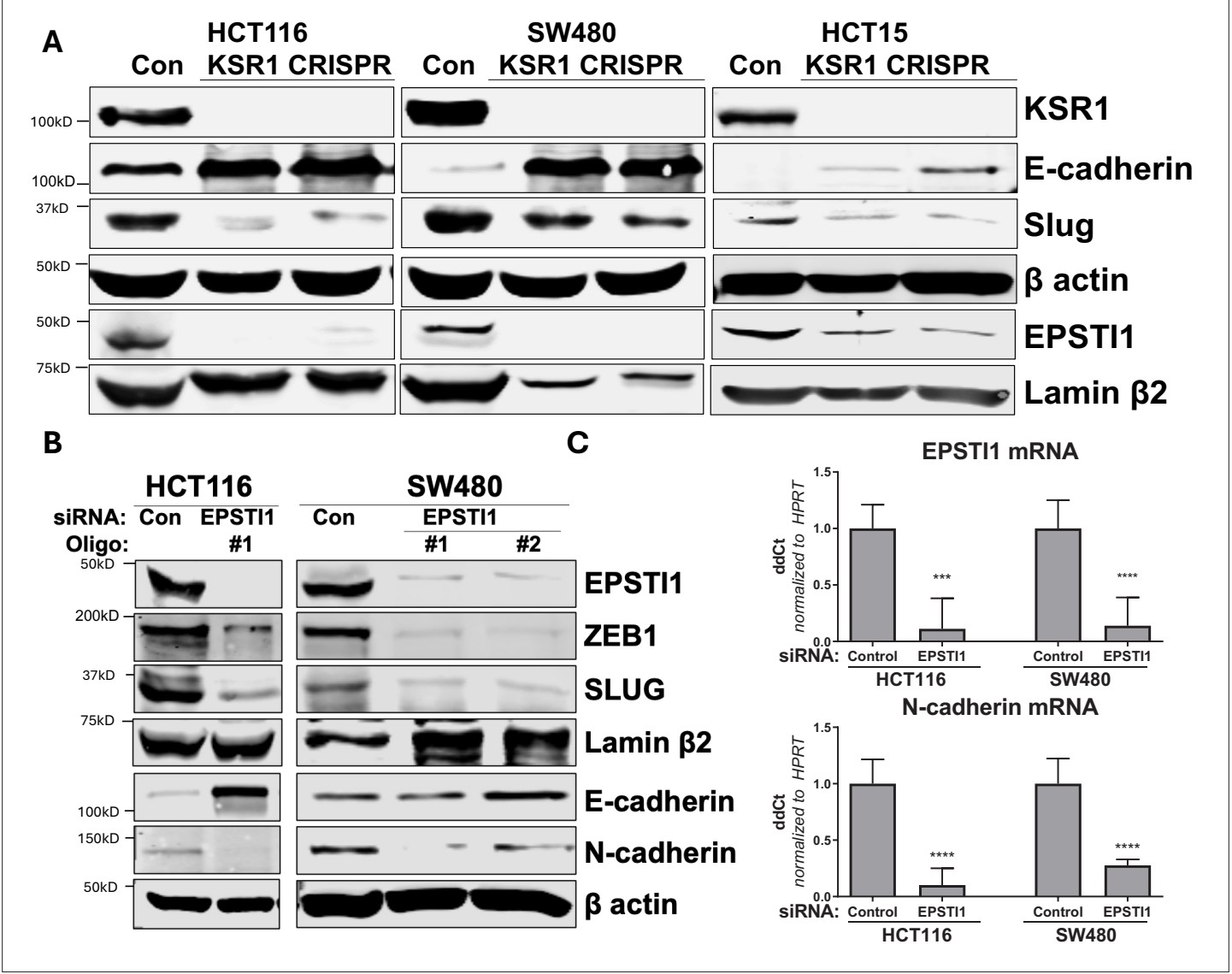

**Figure 5.** KSR1 and EPSTI1 promote cadherin switching. (**A**) Western blot analysis of the cell lysates prepared from control, and two clones of CRISPR-targeted HCT116, SW480, and HCT15 cells (KSR1 CRISPR) for the E-cadherin, Slug, and EPSTI1. (**B**) Western blot of ZEB1, Slug, E-cadherin, and N-cadherin in HCT116 and SW480 cells 72 hr following EPSTI1 knockdown. (**C**) RT-qPCR analysis of EPSTI1 mRNA (upper) and N-cadherin (lower) following knockdown of EPSTI1 for 72 hr in HCT116 and SW480 cells. n = 6; ***, p < 0.001; ****, p < 0.0001. Western blots shown in (**A**) and (**B**) and qPCR shown in (**C**) are representative of at least three independent experiments.

The online version of this article includes the following figure supplement(s) for figure 5:

**Figure supplement 1.** KSR1 and EPSTI1 promote the cadherin switch.

EPSTI1 expression in KSR1 knockout cells had no significant effect on cell proliferation for 24 hr in HCT116 and SW480 cells (**Figure 6—figure supplement 1A, B**), EPSTI1 expression increased the number of invading cells in that period over 50 % in HCT116 and over 70 % in SW480 cells (**Figure 6D**). Although, EPSTI1 expression has a significant effect on cell proliferation over 7 days compared to KSR1 knockout cells in HCT116 and SW480 cells, EPSTI1 expression rescued migratory potential by over 60 % in KSR1-depleted HCT116 and SW480 cells within 24 hr (**Figure 6C**). These data reveal that disabling the cadherin switch and inhibition of cell invasion by KSR1 disruption interrupts EPSTI1 translation, highlighting the pivotal role of this pathway for the induction of EMT-like phenotype in CRC cells.

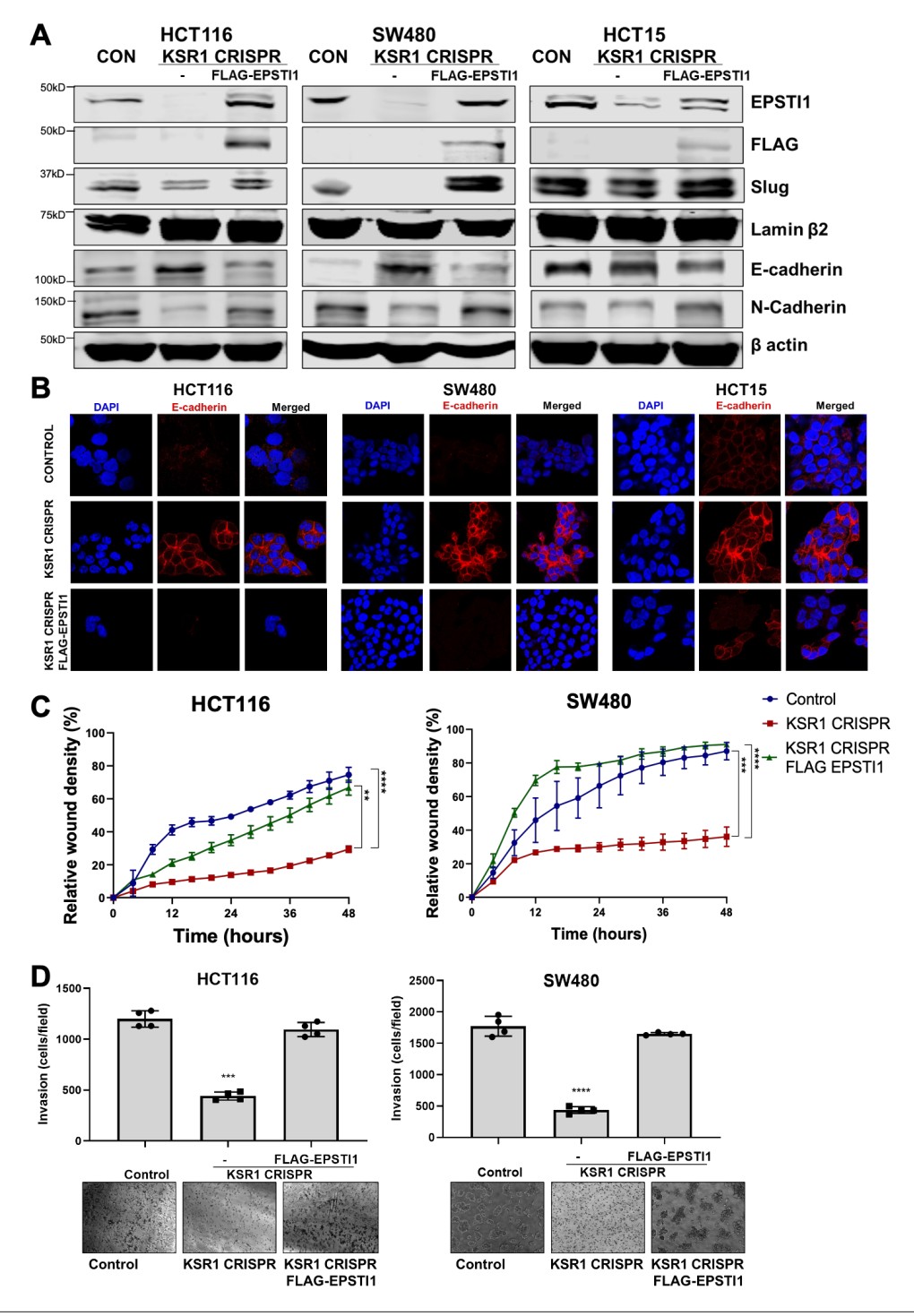

**Figure 6.** EPSTI1 rescues cadherin switching and invasive behavior to KSR1 knockout cells. (**A**) EPSTI1 protein expression was assessed by western blotting in control, KSR1-targeted (KSR1 CRISPR) HCT116, SW480, and HCT15 cells with and without EPSTI1 (FLAG-EPSTI1) expression. Cells were lysed and probed for Slug, E-cadherin, N-cadherin, Lamin β2, and β actin. (**B**) Immunofluorescence staining for E-cadherin (Red) and DAPI (blue) in control or KSR1-targeted (KSR1 CRISPR) HCT116, SW480, and HCT15 cells with and without EPSTI1 (FLAG-EPSTI1) expression. (**C**) Control, CRIPSR- targeted (KSR1-CRISPR), and CRISPR-targeted HCT116 and SW480 cells expressing EPSTI1 (KSR1 CRISPR+ FLAG-EPSTI1) were subjected to the 96-well IncuCyte scratch wound assay. The graph represents the time kinetics of percent wound density, calculated by IncuCyte ZOOM software, shown

*Figure 6 continued on next page*

*Figure 6 continued*

as mean ± SD, n = 12; **, p < 0.005; ***, p < 0.001; ****, p < 0.0001. Matched results were analyzed for statistical significance using one-way ANOVA with Dunnett's posttest for multiple comparisons. (**D**) Control, CRISPR-targeted (KSR1 CRISPR), and CRISPR-targeted HCT116 and SW480 cells expressing EPSTI1 (KSR1 CRISPR+ FLAG-EPSTI1) were subjected to Transwell migration assay through Matrigel . The number of invaded cells per field were counted, (n = 4); ****, p < 0.0001. Representative microscopic images of the respective cells following invasion through Matrigel are shown.

The online version of this article includes the following figure supplement(s) for figure 6:

**Figure supplement 1.** EPSTI1 rescues cadherin switching and invasive behavior to KSR1 knockout cells.

## EPSTI1 re-expression reverses the KSR1-dependent growth inhibition and N-cadherin gene expression

Knockdown of EPSTI1 in HCT116 and SW480, decreased N-cadherin mRNA expression 50 % (*Figure 5C*). Upon KSR1 depletion, N-cadherin mRNA decreased 32 % in HCT116% and 89% in SW480 cells (*Figure 7A*). Ectopic expression of EPSTI1 in these cells restored the N-cadherin mRNA expression to levels observed in control SW480 cells, while in HCT116 KSR1 KO, forced EPSTI1 expression increased N-cadherin mRNA levels threefold above that seen in control HCT116 cells (*Figure 7A*). We tested the effect of ectopic expression of EPSTI1 on invasion in non-transformed HCECs. We stably expressed MSCV-IRES-GFP or MSCV-IRES-EPSTI1-GFP in HCECs and subjected the cells to Transwell invasion assay through Matrigel (*Figure 7B*, top), and observed that EPSTI1 alone was sufficient to dramatically induce the expression of N-cadherin and double the invasive activity of HCECs (*Figure 7B*). These data indicate that EPSTI1 mediates the expression of N-cadherin to promote invasive behavior in non-transformed colon epithelial cells and colon cancer cells.

The E- to N-cadherin switch promotes cancer cell survival following the loss of cell adhesion to the extracellular matrix (*Derksen et al., 2006*; *Onder et al., 2008*). KSR1 also promotes CRC cell survival when detached from a solid substrate (*Fisher et al., 2015*; *McCall et al., 2016*). To determine the extent to which EPSTI1 expression was sufficient to restore CRC cell viability in the absence of KSR1, we grew cells under anchorage-independent conditions either on Poly-(HEMA) (*Figure 7C*) or on soft agar (*Figure 7D*) following forced expression of EPSTI1 in HCT116, HCT15, and SW480 cells lacking KSR1. Anchorage-independent viability was measured over three days on poly-(HEMA) coated plates. Compared to control HCT116 and HCT15 cells, viability decreased approximately 75 % in cells lacking KSR1. Ectopic expression of EPSTI1 restored viability to approximately 50 % of control levels in both cell lines (*Figure 7C*). Similar to our previous findings (*Fisher et al., 2015*; *Kortum et al., 2006*), KSR1 disruption hampered the ability of Ras transformed cells to form colonies on soft agar, the number of colonies formed in HCT116 and SW480 cells dramatically decreased by 75 % in the absence of KSR1. Forced expression of EPSTI1 was sufficient to reverse the suppression of colony formation caused by KSR1 disruption to levels observed in control HCT116 and SW480 cells (*Figure 7D*). These results show that despite the absence of KSR1 to maintain and support cell growth, ectopic EPSTI1 expression was able to maintain anchorage-independent viability in CRC cells.

## Discussion

Persistent oncogenic reprogramming of transcription and translation during EMT grants migratory and invasive properties to tumor cells (*Dongre and Weinberg, 2019*; *Nieto et al., 2016*). Multiple studies have established a relationship between oncogenic Ras-mediated ERK signaling and EMT, either through Ras or its downstream effector signaling pathways activating EMT-TFs (*Shin et al., 2010*; *Andreolas et al., 2008*; *Liu et al., 2014*; *Wong et al., 2013*; *Wang et al., 2010*; *Lemieux et al., 2009*; *Diesch et al., 2014*). Silencing of Erbin, a tumor suppresser known to disrupt KSR1-RAF1 interaction, promoted cell migration and invasion of colon cancer cells, but did not identify the mechanism on how KSR1-dependent MAPK signaling affected EMT (*Stevens et al., 2018*). Mediators of EMT- like phenotype activate cap-dependent translation initiation have been associated with increased aggressiveness and metastases of cancer cells, and we have shown that KSR1 can affect translation initiation (*McCall et al., 2016*; *Jechlinger et al., 2003*; *Waerner et al., 2006*; *Prakash et al., 2019*).

Our observations establish the novel role of the scaffold protein KSR1 promoting the preferential translation of an EMT-related gene, *EPSTI1*, and outline a mechanism for KSR1-dependent stimulation

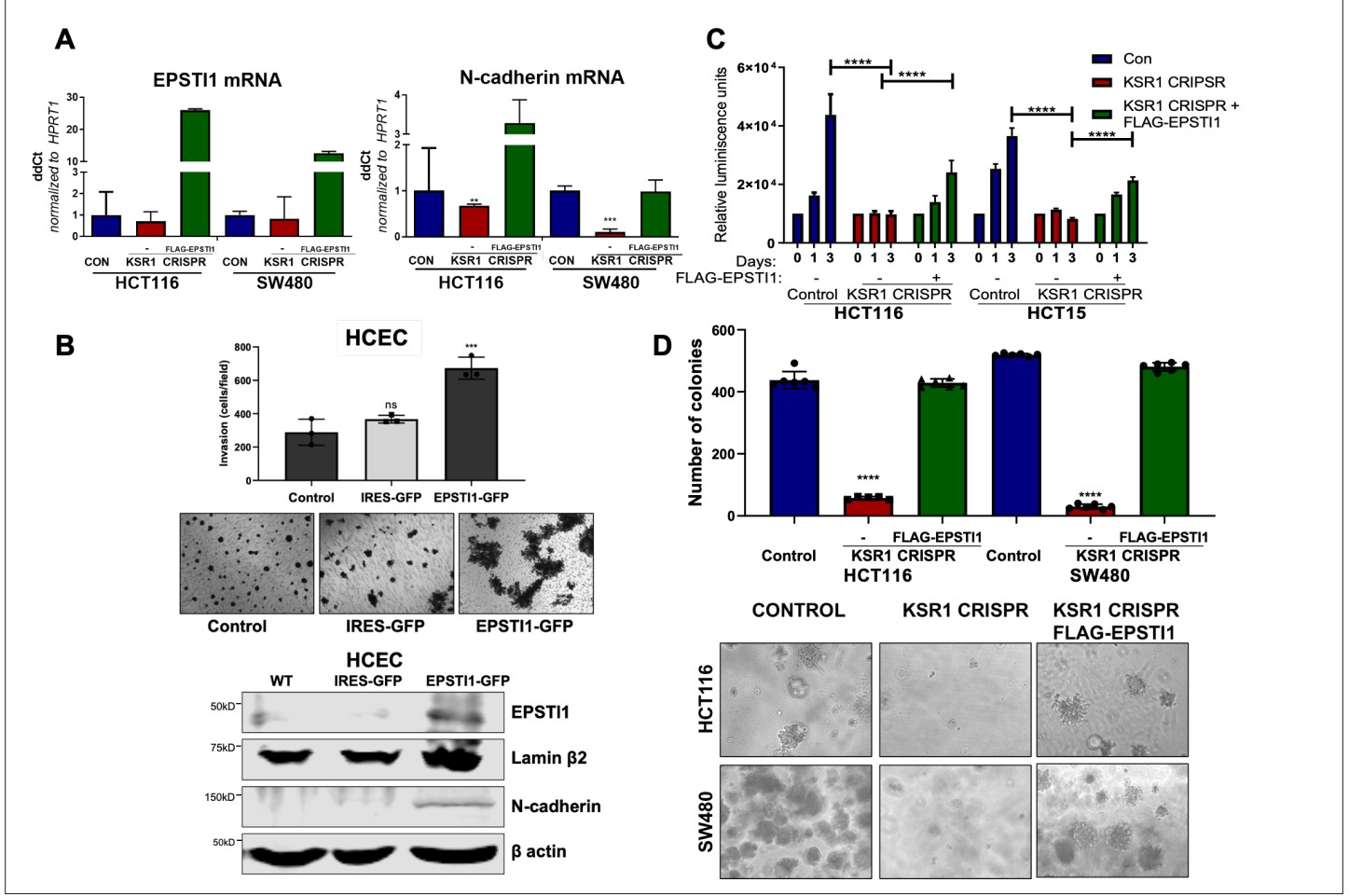

**Figure 7.** EPSTI1 expression in KSR1 KO CRC cells and HCEC cells induces N-cadherin expression and restores anchorage-independent growth in CRC cells. (**A**) RT-qPCR analysis of EPSTI1 mRNA (left) and N- cadherin (right) in HCT116 and SW480 cells following KSR1 disruption with and without expression of EPSTI1 (FLAG-EPSTI1) in KSR1 KO cells. (n = 3), **, $p < 0.01$***, $p < 0.001$ (**B**) (Top panels) Wild-type (WT) HCEC, HCECs transfected with MSCV-IRES-GFP (IRES-GFP) or MSCV-FLAG-EPSTI1-IRES-GFP (EPSTI1-GFP) were subjected to Transwell migration assay through Matrigel . The number of invaded cells per field were counted, (n = 4); ***; $p < 0.001$. Representative microscopic images of the respective cells following invasion through Matrigel are shown. (Bottom panels) Western blot analysis of EPSTI1 and N-cadherin from the cell lysates prepared from Wild-type (WT) HCEC, HCECs transfected with MSCV-IRES-GFP (IRES-GFP) or MSCV-FLAG-EPSTI1-IRES-GFP (EPSTI1-GFP). (**C**) KSR1 KO HCT116 and HCT15 cell viability (CellTiter-Glo) on poly-(HEMA)-coated plates at the indicated days with or without EPSTI1 (KSR1 CRISPR + EPSTI1) expression. The data are shown as relative luminescence units mean ± SD, (n = 6); ****; $p < 0.0001$. The data were analyzed for statistical significance by one-way ANOVA followed by t-test. (**D**) Quantification of anchorage-independent colonies formed by KSR1 knockout HCT116 and SW480 cells with and without EPSTI1 expression (KSR1 CRISPR+ FLAG-EPSTI1) after plating in soft agar. Representative photomicrographs of colonies from each cell line are shown. The data are illustrated as the number of colonies present after 2 weeks, (n = 6) mean ± SD. ****, $p < 0.0001$. Data were analyzed for statistical significance one-way ANOVA followed by t-test.

of phenotypic plasticity. Using gene-expression analysis of the polysome-bound mRNA, we discovered KSR1 and ERK increase the translational efficiency of *EPSTI1* mRNA. EPSTI1 mediates KSR1-dependent motility, invasion, and anchorage-independent growth coincident with its suppression of EMT-TF, Slug, elevating E-cadherin expression. EPSTI1 knockdown also decreased the expression of N-cadherin mRNA and protein. In the absence of KSR1, ectopic expression of EPSTI1 was sufficient to suppress E-cadherin expression, stimulate N-cadherin expression and enhance motility and invasive behavior, this invasive behavior was also induced in non-transformed colon cells. These data demonstrate that a KSR1- and ERK-regulated component is critical to the execution of the transcriptional program that drives interconversion between epithelial and mesenchymal phenotypes. These studies of post-transcriptional regulation and mRNA translation reveal the importance of expanding beyond gene expression analysis for detecting mechanisms underlying epithelial plasticity and tumorigenicity.

The association of EPSTI1 with tumor metastatic potential is supported by observations that *EPSTI1* is highly upregulated in invasive breast cancer tissues and suggested the role of EPSTI1 in promoting metastasis, tumorsphere formation, and stemness (*Nielsen et al., 2002*; *Li et al., 2014*; *de Neergaard et al., 2010*). Although the aberrant expression of EPSTI1 in breast cancer cells is well-established, there is little indication in the literature on the role of EPSTI1 to induce EMT, cancer invasion, and metastasis. The association of EPSTI1 induction of invasion in breast cancer cells was attributed to the increased expression of *Slug* and *Twist* mRNA and increased expression of fibronectin and α2β1 integrins (*de Neergaard et al., 2010*). Another study suggested the interaction of EPSTI1 with valosin-containing protein (VCP) and the subsequent activation of NF-κB signaling contributed to the increased tumor invasion and metastasis (*Li et al., 2014*). Future studies should evaluate the potential of EPSTI1 to directly affect N-cadherin and EMT-TF expression, assess the role of NF-κB signaling in EPSTI1-dependent CRC cell EMT and evaluate the potential of EPSTI1 to contribute to invasion and metastasis in vivo.

Determining how KSR1- and ERK-dependent signaling promotes EPSTI1 translation should yield novel mechanisms underlying tumor cell metastatic behavior. We show that EPSTI1 mRNA is unchanged upon KSR1 disruption or ERK inhibition (*Figures 1E and 2D*) and KSR1 does not contribute to UPS-mediated protein degradation (*Figure 1—figure supplement 2E*), suggesting that KSR1 regulates EPSTI1 through post-transcriptional modifications enhancing its preferential loading onto the polysomes. Differential mRNA splicing is implicated in EMT-related processes and splicing regulatory factors have been implicated in the motility and invasive behavior of tumor cells (*Park et al., 2019*; *Pradella et al., 2017*). One possibility is that KSR1 signaling promotes the splicing of EPSTI1 that promotes it's the preferential translational contributing to increased motility and invasion.

Upon removal of KSR1 or EPSTI1, the tumor cells switch back from highly migratory and invasive EMT-like state to the epithelial state. However, the invasive property is not completely lost in KSR1/EPSTI1 disruption (*Figure 4B*), which could be attributed to other mesenchymal markers retained in the cells, such as vimentin (*Figure 5—figure supplement 1A*). Investigating other EMT-related mRNAs that are preferentially translated in response to KSR1-scaffolded ERK signaling may reveal additional mRNAs that make previously unappreciated contributions to cell migration, invasion, and EMT. Constitutive KSR1 or EPSTI1 knockout yields developmentally normal mice (*Lozano et al., 2003*; *Nguyen et al., 2002*; *Kim et al., 2018*). While KSR1 or EPSTI1 may not be essential to EMT during normal development, they may play a role in other EMT-dependent events such as wound healing where cells collectively migrate, differentiate, and re-epithelialize keratinocytes around and/or within the damaged site. If their role in EMT is exclusive to tumor cells it will reveal a key vulnerability for therapeutic evaluation. Further characterization of KSR1, EPSTI1 and the additional effectors repurposed by dysregulated translation in CRC should reveal additional novel mechanisms critical to CRC tumor survival and progression.

# Materials and methods

## Key resources table

| Reagent type (species) or resource | Designation | Source or reference | Identifiers | Additional information |
|---|---|---|---|---|
| Cell line (*Homo sapiens*) | Colorectal carcinoma, epithelial | ATCC | HCT116 (ATCC, Cat# CCL-247, RRID:CVCL_0291) | |
| Cell line (*Homo sapiens*) | Colorectal carcinoma, epithelial | ATCC | HCT15 (ATCC, Cat# CCL-225, RRID:CVCL_0292) | |
| Cell line (*Homo sapiens*) | Colorectal adenocarcinoma, epithelial | ATCC | SW480 (ATCC, Cat# CCL-228, RRID:CVCL_0546) | |
| Cell line (*Homo sapiens*) | Immortalized colon epithelial | Obtained from Dr. Jerry Shay | HCEC | |
| Cell line (*Homo sapiens*) | Kidney; epithelial fibroblast (fetus) | ATCC | HEK-293T (ATCC Cat# CRL-3216, RRID:CVCL_0063) | |
| Cell line (*Homo sapiens*) | Kidney; epithelial fibroblast (fetus) | Obtained from Rob Kortum | Phoenix-GP | Available at ATCC (Cat# CRL-321) |

*Continued on next page*

*Continued*

| Reagent type (species) or resource | Designation | Source or reference | Identifiers | Additional information |
|---|---|---|---|---|
| Transfected construct (*Homo sapiens*) | siRNA to non-targeting control | Dharmacon | Cat# D-001810-01-20 | UGGUUUACAUGUCGACUAA |
| Transfected construct (*Homo sapiens*) | siRNA to EPSTI1 | Dharmacon | Cat# 015094-09-0020 | GAACAGAGCUAAACCGGUU |
| Transfected construct (*Homo sapiens*) | siRNA to EPSTI1 | Dharmacon | Cat# 015094-12-0020 | UCUGGAGGCUGUUGGAAUA |
| Transfected construct (*Homo sapiens*) | Con shRNA#1 | *Fisher et al., 2015* | pLKO.1 MC1 puro | CAACAAGATGAAGAGCACCAA |
| Transfected construct (*Homo sapiens*) | KSR1 shRNA#1 | *Fisher et al., 2015* | pLKO.1 KSR.1 puro | GTGCCAGAAGAGCATGATTTT |
| Transfected construct (*Homo sapiens*) | KSR1 shRNA#2 | *Fisher et al., 2015* | pLKO.1 KSR.2 puro | GCTGTTCAAGAAAGAGGTGAT |
| Transfected construct (*Homo sapiens*) | CON sgRNA#1 | This paper | pCAG-SpCas9-GFP-U6-gNC1 | GTATTACTGATATTGGTGGG |
| Transfected construct (*Homo sapiens*) | KSR1 sgRNA#1 | This paper | pCAG-SpCas9-GFP-U6-gCR1.1 | TTGGATGCGCGGCGGGAAAG |
| Transfected construct (*Homo sapiens*) | KSR1 sgRNA#2 | This paper | pCAG-SpCas9-GFP-U6-gCR1.2 | CTGACACGGAGATGGAGCGT |
| Recombinant DNA reagent | FLAG-KSR1 (plasmid) | *Fisher et al., 2015* | MSCV-KSR1-IRES-GFP | |
| Recombinant DNA reagent | FLAG-EPSTI1 (plasmid) | This paper | MSCV-FLAG-EPSTI1-IRES-GFP | MGC Human EPSTI1 Sequence-Verified cDNA (Cat# MHS6278-202832484) cloned into MSCV-IRES-GFP construct |
| Recombinant DNA reagent | N-cad OE (plasmid) | *Gift from Dr. Keith Johnson* | N-cadherin-mGFP | |
| Sequence-based reagent | EPSTI1 (PCR primer) | IDT | Cat# Hs.PT.58.50471678 | Forward primer 5'-GTGAATTACTGGAACTGAAACGG-3' Reverse primer 5' TCCAACAGCCTCCAGATTG 3' Tm 55 °C, Exon Location 10–11 |
| Sequence-based reagent | N-cadherin (PCR primer) | IDT | Cat# Hs.PT.58.26024443 | Forward primer 5'-GTTTGCCAGTGTGACTCCA-3' Reverse primer 5'-CATACCACAAACATCAGCACAAG-3' Tm 55 °C, Exon Location 13–14 |
| Sequence-based reagent | HPRT1 (PCR primer) | IDT | Cat# Hs.PT.58v.45621572 | Forward Primer: 5' GTATTCATTATAGTCAAGGGCATATCC 3' Reverse Primer: 5'AGATGGTCAAGGTCGCAAG 3' Tm 60 °C, Exon Location 8–9 |
| Sequence-based reagent | ZEB1 (PCR primer) | IDT | Cat# Hs.PT.58.39178574 | Forward primer 5'-GAGGAGCAGTGAAAGAGAAGG-3' Reverse primer 5'-TACTGTACATCCTGCTTCATCTG-3' Tm 60 °C, Exon Location 3–5 |
| Sequence-based reagent | SLUG (PCR primer) | IDT | Cat# Hs.PT.58.50471678 | Forward primer 5'-AGGACACATTAGAACTCACACG-3' Reverse primer 5'-CAGATGAGCCCTCAGATTTGAC-3' Tm 55 °C, Exon Location 2–3 |
| Antibody | Anti-KSR1, Rabbit polyclonal | Abcam | Cat# ab68483 | WB (1:1000) |
| Antibody | Anti-EPSTI1, Rabbit polyclonal | Proteintech | Cat# 11627–1-AP, RRID:AB_2877786 | WB (1:1000) |
| | | *Gift from Dr. Keith Johnson* | Cat# 13A9 | WB (1:20) |
| Antibody | Anti-N-cadherin | Cell Signaling | Cat# 13116, RRID:AB_2687616 | WB (1:1000) |
| | | *Gift from Dr. Keith Johnson* | Cat# 4A2 | WB (1:10) IF (1:1) |
| Antibody | Anti-E-cadherin | Cell Signaling | Cat# 3195, RRID:AB_2291471 | WB (1:1000) |
| Antibody | Anti-Slug, Rabbit monoclonal | Cell Signaling Technology | Cat# 9585, RRID:AB_2239535 | WB (1:1000) |

*Continued on next page*

*Continued*

| Reagent type (species) or resource | Designation | Source or reference | Identifiers | Additional information |
|---|---|---|---|---|
| Antibody | Anti-Lamin β2, Rabbit monoclonal | Abclonal | Cat# A6483, RRID:AB_2767083 | WB (1:2000) |
| Antibody | Anti-β actin, Mouse monoclonal | Santa Cruz | Cat# 47778, RRID:AB_2714189 | WB (1:2000) |
| Antibody | Anti-phospho RSK S380, Rabbit polyclonal | Cell Signaling Technology | Cat# 9341, RRID:AB_330753 | WB (1:500) |
| Antibody | Anti-Total RSK, Rabbit monoclonal | Cell Signaling Technology | Cat# 9355, RRID:AB_659900 | WB (1:1000) |
| Antibody | Anti-phospho p70S6K T389, Rabbit polyclonal | Cell Signaling Technology | Cat# 9206, RRID:AB_2285392 | WB (1:500) |
| Antibody | Anti-total p70S6K, Rabbit polyclonal | Cell Signaling Technology | Cat# 9202, RRID:AB_331676 | WB (1:1000) |
| Antibody | Anti-SNAIL, Rabbit monoclonal | Cell Signaling Technology | Cat# 3879, RRID:AB_2255011 | WB (1:1000) |
| Antibody | Anti-Vimentin, Rabbit monoclonal | Cell Signaling Technology | Cat# 5741, RRID:AB_10695459 | WB (1:1000) |

## Cell culture

Colorectal cancer cell lines HCT116, HCT15, and SW480 were acquired from American Type Culture Collection (ATCC). The cells were cultured in Dulbecco's modified Eagle's medium (DMEM) containing high glucose with 10 % fetal bovine serum (FBS) and grown at 37 °C with ambient $O_2$ and 5 % $CO_2$. Cells were routinely tested for mycoplasma. No further authentication of cell lines was performed by the authors. Non-transformed immortalized human colon epithelial cell line (HCEC) was a gift from J. Shay (University of Texas [UT] Southwestern) and were grown and maintained as described previously (*Fisher et al., 2015*; *Roig et al., 2010*). HCECs were grown in a hypoxia chamber with 2% $O_2$ and 5% $CO_2$ at 37°C in four parts DMEM to 1 part medium 199 (Sigma-Aldrich #M4530) with 2% cosmic calf serum (GE Healthcare, #SH30087.03), 25 ng/mL EGF (R&D, Minneapolis, MN #236-EG), 1 μg/mL hydrocortisone (Sigma-Aldrich, #H0888), 10 μg/mL insulin (Sigma-Aldrich, #I550), 2 μg/mL transferrin (Sigma-Aldrich, #T1428), 5 nM sodium selenite (Sigma-Aldrich #S5261), and 50 μg/mL gentamicin sulfate (Gibco #15750–060) as described previously (*Fisher et al., 2015*). Normal and quadruple mutant AKPS (APC $^{KO}$/KRAS$^{G12D}$/P53 $^{KO}$/SMAD4$^{KO}$) tumor colon organoids obtained from the Living Organoid Biobank housed by Dr. Hans Clevers and cultured as described previously (*Drost et al., 2015*; *van de Wetering et al., 2015*). The normal organoids were cultured in medium containing advanced DMEM/F12 (Invitrogen #12634) with 50 % WNT conditioned media (produced using stably transfected L cells), 20 % R-spondin1, 10 % Noggin, 1 X B27 (Invitrogen #17504–044), 10 mM nicotinamide (Sigma-Aldrich #N0636), 1.25 mM N-acetylcysteine (Sigma-Aldrich #A9165-5G), 50 ng/mL EGF (Invitrogen #PMG8043), 5000 nM TGF-b type I receptor inhibitor A83-01 (Tocris #2939), 10 nM Prostaglandin E2 (Tocris #2296), 3 μM p38 inhibitor SB202190 (Sigma-Aldrich #S7067), and 100 μg/mL Primocin (Invivogen #ant-pm-1). The quadruple mutant AKPS organoids were grown in media lacking WNT conditioned media, R-spondin 1, noggin and EGF and containing 10 μM nutlin-3 (Sigma #675576-98-4).

## RNA interference

Approximately 500,000 cells were transfected using a final concentration of 20 nM EPSTI1 (J-015094-09-0020 and J-015094-12-0020) or non-targeting (D-001810-01-20 and D-001810-02-20) ON-TARGET-plus siRNAs from GE Healthcare Dharmacon using 20 μL of Lipofectamine RNAiMAX (ThermoFisher #13778–150) and 500 μL OptiMEM (ThermoFisher #31985070). Cells were incubated for 72 hr before further analysis.

## Generation of KSR1 shRNA knockdown and KSR1 CRISPR/Cas9 knockout cell lines

A lentiviral pLKO.1-puro constructs targeting KSR1, and non-targeting control were transfected into HEK-293T cells using trans-lentiviral packaging system (ThermoFisher Scientific). The virus was collected, and the medium was replaced 48 hr post transfection. HCT116 and HCT15 cells were infected with virus with 8 µg/mL of Polybrene for several days. The population of cells with depleted KSR1 was selected with 10 µg/mL puromycin. The KSR1 knockdown was confirmed via western blotting.

pCAG-SpCas9-GFP-U6-gRNA was a gift from Jizhong Zou (Addgene plasmid #79144), KSR1 sgRNA and non-targeting control sgRNA was cloned into the pCas9 vector. Both the non-targeting control and sgKSR1 were transfected into HCT116, HCT15, and SW480 cells using PEI transfection as described previously (*Longo et al., 2013*). The GFP-positive cells were sorted 48 hr post transfection, and colonies were picked by placing sterile glass rings around individual colonies.

MSCV-IRES-GFP, MSCV-IRES-KSR1-GFP, MSCV-IRES-FLAG-EPSTI1, and N-cadherin mGFP constructs were transfected into Phoenix GP cells using trans-lentiviral packaging system (ThermoFisher Scientific). The virus was collected, and the medium was replaced 48 hours post transfection. HCECs/ KSR1-CRISPR HCT116, HCT15, and SW480 cells were infected with virus with 8 µg/mL of Polybrene for 96 hours. The population of cells with KSR1 expression was selected following GFP sorting using fluorescence-activated cell sorting (FACS). The KSR1/EPSTI1 expression was confirmed via western blotting.

## Cell lysis and western blot analysis

Whole cell lysate was extracted in radioimmunoprecipitation assay (RIPA) buffer containing 50 mM Tris-HCl, 1% NP-40, 0.5 % Na deoxycholate, 0.1 % Na dodecyl sulfate, 150 mM NaCl, 2 mM EDTA, 2 mM EGTA, and 1 X protease and phosphatase inhibitor cocktail (Halt, ThermoFisher Scientific #78440). Cytoplasmic and nuclear fractionation was performed using NE-PER Nuclear and Cytoplasmic Extraction Reagents (ThermoFisher Scientific #PI78835). The estimation of protein concentration was done using BCA protein assay (Promega #PI-23222, PI-23224). Samples were diluted using 1 X sample buffer (4 X stock, LI-COR #928–40004) with 100 mM dithiothreitol (DTT) (10 X stock, 1 mM, Sigma #D9779-5G). The protein was separated using 8–12% SDS-PAGE and transferred to nitrocellulose membrane. The membrane was blocked with Odyssey TBS blocking buffer (LICOR-Biosciences #927–50003) for 45 min at room temperature, then incubated with primary antibodies (*Key Resources Table*) at least overnight at 4°C. IRDye 800CW and 680RD secondary antibodies (LI-COR Biosciences # 926–32211, # 926–68072) were diluted 1:10,000 in 0.1% TBS-Tween and imaged on the Odyssey Classic Scanner (LI-COR Biosciences).

## Polysome profiling

Cells were treated with 100 µg/mL cycloheximide (Sigma #C4859) on ice in PBS for 10 min. The cells were lysed with 10 mM HEPES, 100 mM KCL, 5 mM MgCl$_2$, 100 µg/mL cycloheximide, 2 mM DTT, 1 % Triton-X100, 2.5 µl RNaseOUT (ThermoFisher Scientific #10777019). The lysates were cleared by centrifugation for 10 min at 13,200 rpm at 4°C. Approximately 200 µL of the total RNA was collected in a new RNAse-free microcentrifuge tube and the remaining supernatant was loaded onto a 15–45% sucrose gradient. The samples were spun at 37,500 rpm for 2 hr at 4°C in SW55Ti Beckman ultracentrifuge and separated on a gradient fractionation system to resolve the polysomes. Polysome profiles were identified at 260 nM using an absorbance detector. Gradient fractions were collected dropwise at 0.75 mL/min. For RNAseq, the total RNA and RNA pooled from the polysome fraction (fractions 6–9) of three sets of independently isolated cells was isolated using RNAzol (Molecular Research Centre #RN 190) according to the manufacture's protocol. RNA purity was evaluated by the UNMC DNA Sequencing Core using a BioAnalyzer.

## RNA-sequencing and analysis

RNA sequencing (RNA seq) was conducted by the UNMC DNA Sequencing Core. For RNA-seq, RNA was purified from three biological replicates of total and polysome-bound RNA from HCT116 and HCT15, control and KSR1 knockdown cells as previously described. Stranded RNA sequencing libraries were prepared as per manufactures' protocol using TrueSeq mRNA protocol kit (Illumina)

and 500 ng of the total RNA was used for each of the samples. Purified libraries were pooled at a 0.9 pM concentration and sequenced on an Illumina NextSeq550 instrument, using a 75 SR High-output flow cell, to obtain approximately 45 million single-end reads per sample. NGS short reads from RNA-seq experiments was downloaded from the HiSeq2500 server in FASTQ format. FastQC (http://www.bioinformatics.babraham.ac.uk/projects/fastqc/) was used to perform quality control checks on the *fastq* files that contain the raw short reads from sequencing. The reads were then mapped to the *Homo sapiens* (human) reference genome assembly GRCh38 (hg38) using STAR v2.7 alignment. The `--quantMode GeneCounts` option in STAR 2.7 (*Dobin et al., 2013*) was used to obtain the HTSeq counts per gene. Gencode v32 Gene Transfer Format (GTF) was used for the transcript/gene annotations. The output files were combined into a matrix using R. The gene counts were further used as input for downstream analysis using Anota2seq. The high-throughput sequencing data have been deposited in the Gene Expression Omnibus (GEO) database, http://www.ncbi.nlm.nih.gov/geo (accession no. GSE164492).

## Translational efficiency

The altered levels of total mRNA can impact the changes in the pool of polysome-bound mRNA, leading to a spurious calculation translational efficiency (TE). Anota2seq (*Oertlin et al., 2019*) allows the quantification of actual changes in TE. TE was calculated using the R Bioconductor anota2Seq package for the HTSeq counts by first removing genes that did not contain expression values in more than 10% of the samples. 16,023 genes remained after this step. TMM normalization was further performed prior to log2 counts per million computation (CPM) using the voom function of the limma package using the anota2seqDataSetFromMatrix function (with parameters datatype = "RNAseq", normalize = TRUE, transformation = "TMM-log2"). TE was calculated using the 2 × 2 factorial design model for the two cell lines (HCT116 and HCT15). Genes were considered significantly regulated at Adjusted p-value < 0.05 when passing filtering criteria (parameters for anota2seqSelSigGenes function) using Random variance Model [useRVM = TRUE], [selDeltaPT > log2(1.2)], [minSlopeTranslation >−1], [maxSlopeTranslation <2], [selDeltaTP> log2(1.2)], [minSlopeBuffering >−2] and [maxSlopeBuffering <1], [selDeltaP> log2 (1)], [selDetaT > log2 (1)]. The scatterplots were obtained using the anota2seqPlotFC function. The heatmaps were generated using the TE values for the two cell lines using the R Bioconductor ComplexHeatmap package.

## Anchorage-independent growth [poly-(HEMA)] assay

Poly-(HEMA) stock solution (10 mg/mL) was prepared by dissolving poly-(HEMA) (Sigma #3932–25 G) in 95% ethanol at 37°C until fully dissolved (overnight). Ninety-six-well optical bottom plates (Thermo Scientific Nunc #165305) were coated in 200 µL of poly-(HEMA) solution and allowing it to evaporate. Cells were plated in complete growth medium of the poly-(HEMA) coated plates at a concentration of 10,000 cells/ 100 µL. Cell viability was measured at the indicated time points by the addition of Cell-Titer-Glo 2.0 reagent (Promega #G9242) and luminescence was measured (POLARstar Optima plate reader) according to the manufacturer's protocol.

## Anchorage-independent growth (soft agar) assay

A total of 6000 cells were seeded in 1.6 % NuSieve Agarose (Lonza #50081) to assess anchorage-independent growth according to the protocol of *Fisher et al., 2015*. Colonies greater than 100 µm in diameter from six replicates per sample were counted, representative photomicrographs were taken after 10–14 days of incubation at 37°C and 5 % CO2.

## RT-qPCR

Cells were harvested using 1 mL TRIzol (ThermoFisher Scientific #15596026) and RNA extraction was performed using RNeasy spin columns (Qiagen #74104). RNA was eluted with nuclease-free water. The RNA was quantified using a NanoDrop 2000 (Thermo Scientific) and Reverse Transcription (RT) was performed with 2 µg RNA per 40 µl reaction mixture using iScript Reverse Transcription Supermix (Bio-Rad #170–8891). RT-qPCR was performed using primers antibodies (*Key Resources Table*), and all targets were amplified using SsoAdvanced Universal SYBR green Supermix (Bio-Rad #1725271) with 40 cycles on a QuantStudio 3 (ThermoFisher Scientific). The analysis was performed using $2^{-\Delta\Delta C_T}$ method (*Schmittgen and Livak, 2008*). For polysome gradients, the RNA levels were quantified from

the cDNA using the standard curve method, summed across all fractions (*Kortum and Lewis, 2004*; *Nguyen et al., 2002*; *Fisher et al., 2011*; *Fisher et al., 2015*; *Morrison et al., 2009*; *Rao et al., 2020*) and presented as a percentage of the total fractions.

### Cell migration (scratch-test) assay

An in vitro scratch test was performed with the IncuCyte Zoom according to the manufacturer's instructions. Approximately 35,000 cells were seeded onto a 96-well ImageLock plates (Essen BioScience #4379) and grown to 90–95% confluency. The scratches were created using WoundMaker (Essen BioScience #4563) in all the wells, after which the cells were washed with 1 x PBS, and media without containing serum was replaced. Images of the cells were obtained every 20 min for a total duration of 72 hr using IncuCyte Kinetic Live Cell Imaging System (Essen BioScience) and analyzed using the IncuCyte Zoom software (Essen BioScience). IncuCyte Software was used to calculate the relative wound density metric to quantify the cell migration over time. The metric is designed to be zero at t = 0% and 100% when cell density inside the wound is the same as the cell density outside the initial wound, thus, allowing to experimentally quantify the effects of cell migration separate from changes that occurs as result of cell proliferation.

### Cell invasion (transwell) assay

Transwell inserts (24-well Millicell cell culture, #MCEP24H48) were coated with 50 µL of Matrigel (Corning, # 356234) and allowed to solidify for 15–30 min. Approximately 20,000 stably generated knockout cells, or cells after 48 hr of transfection were plated in serum free media in the upper chamber of transwell insert. Cells were allowed to invade toward 10% serum containing media in the lower chamber for 24 hr, after which cells and gel in the upper chamber was gently removed with a sterile cotton applicator and the cells in the lower side of the insert was fixed with 3.7% formaldehyde for 2 min, permeabilized with 100% methanol for 20 min and stained with Giemsa for 15 min. The numbers of cells were counted using an inverted microscope at ×20 magnification.

### Immunofluorescence assay

Cells were plated on glass coverslips to 70–80% confluence for 48 hr in growth media. Cells were fixed in 1 % formaldehyde diluted in PBS for 15 min. The cells were rinsed three times with PBS for 5 min and coverslips were blocked for 1 hr with 1 X PBS/ 5% goat serum/0.3% Triton X-100 and then incubated with E-cadherin antibody (#4A2) overnight. Cells were washed three times for 5 min with PBS and incubated in anti-mouse IgG Alexa Fluor 555 Conjugate (Cell signaling #4409) at a dilution of 1:500 for 1 hr. Coverslips were rinsed three times for 5 min in PBS and briefly rinsed in distilled water prior to mounting in Prolong Gold Antifade Reagent with DAPI (Cell signaling #8961). All Images were acquired using a Zeiss LSM-780 confocal microscope and processed using ZEISS ZEN 3.2 (blue edition) software.

### Cell growth assay

Cells were transfected with siRNA targeting EPSTI1 or a non-targeting control as previously described. The next day, control, KSR1-CRISPR, KSR1-CRISPR HCT116 and SW480 cells expressing KSR1 or EPSTI1, siControl and siEPSTI1 HCT116 and SW480 cells were counted and approximately $1 \times 10^4$ cells were plated in all wells of a 12-well plate for each condition. The next day, four of the wells from each 12-well plate were harvested and stained with 0.4% trypan blue (Sigma, # T6146-5G) and were then counted and recorded using Countess II automated cell counter (ThermoFisher, #A27977). This procedure was repeated for indicated days and the cells from day 7 were harvested and a western blot analysis was performed to ensure the expression of the target protein was maintained. Cell counts were then graphed in GraphPad.

### Reagents

The ERK inhibitor SCH772984 was purchased from SelleckChem (S7101), Z-Leu-Leu-Leu-al (MG132, S2619) were purchased from Fisher and mTOR inhibitor AZD8055 (HY_10422) MedChem Express.

## Acknowledgements

We thank Dr. Xuan Zhang, and Dr. Kai Fu for their assistance with polysome profiling, Lisa E Humphrey-Brattain for their assistance with the colon organoid culture, the UNMC Advanced Microscopy Core, and the UNMC Cell Analysis Facility. We declare no conflicts of interest. This work was supported by P20 GM121316 (REL), and the Fred & Pamela Buffett Cancer Center Support Grant (P30 CA036727). The funders had no role in the study design, data collection and interpretation, or the decision to submit the work for publication.

## Additional information

### Funding

| Funder | Grant reference number | Author |
|---|---|---|
| National Institutes of Health | P20 GM121316 | Robert E Lewis |
| National Cancer Institute | P30 CA036727 | Robert E Lewis |

The funders had no role in study design, data collection and interpretation, or the decision to submit the work for publication.

### Author contributions
Chaitra Rao, Conceptualization, Data curation, Formal analysis, Investigation, Methodology, Validation, Visualization, Writing – original draft, Writing – review and editing; Danielle E Frodyma, Investigation, Methodology; Siddesh Southekal, Formal analysis, Investigation, Methodology, software; Robert A Svoboda, Data curation, Methodology, resources; Adrian R Black, Investigation, Methodology, resources; Chittibabu Guda, Tomohiro Mizutani, Hans Clevers, Keith R Johnson, resources; Kurt W Fisher, resources, Writing – review and editing; Robert E Lewis, Conceptualization, funding-acquisition, project-administration, resources, supervision, Visualization, Writing – review and editing

### Author ORCIDs
Chaitra Rao http://orcid.org/0000-0002-2834-7458
Robert E Lewis http://orcid.org/0000-0002-3616-2971

### Decision letter and Author response
Decision letter https://doi.org/10.7554/eLife.66608.sa1
Author response https://doi.org/10.7554/eLife.66608.sa2

## Additional files

### Supplementary files
• Supplementary file 1. Genes translationally altered by KSR1. Translational efficiency of mRNAs (58 decreased; 40 increased) upon KSR1 knockdown in HCT116 and HCT15 cells, related to *Figure 1B*.

• Supplementary file 2. KSR1-dependent genes predicted in mesenchymal-up signature identified by GSEA. GSEA for the subset of translationally controlled genes involved in 'Hallmark EMT signature', 'Jechlinger EMT Up', and 'Gotzmann EMT up' in a KSR1-dependent manner.

• Transparent reporting form

• Source data 1. Translational efficiency of mRNAs significantly altered by KSR1 analyzed using Anota2seq.

### Data availability
The high-throughput sequencing data have been deposited in the Gene Expression Omnibus (GEO) database, http://www.ncbi.nlm.nih.gov/geo (accession no. GSE164492).

The following dataset was generated:

| Author(s) | Year | Dataset title | Dataset URL | Database and Identifier |
|---|---|---|---|---|
| Rao C, Southekal S, Lewis RE | 2021 | | https://www.ncbi.nlm.nih.gov/geo/query/acc.cgi?acc=GSE164492 | NCBI Gene Expression Omnibus, GSE164492 |

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
