## [Decision Letter]

**Acceptance summary:**

This paper demonstrates the involvement of Kinase Suppressor of Ras 1 (KSR1), a protein that acts as a scaffold in the mitogen-activated protein kinase (MAPK) signaling cascade, in translational control of the epithelial-to-mesenchymal transition. The analysis is thorough and includes both loss-of-function and gain-of-function studies. This study advances our understanding of cancer development.

**Decision letter after peer review:**

Thank you for submitting your article "KSR1-and ERK-dependent Translational Regulation of the Epithelial-to-Mesenchymal Transition" for consideration by *eLife*. Your article has been reviewed by 3 peer reviewers, and the evaluation has been overseen by a Reviewing Editor and Jonathan Cooper as the Senior Editor. The following individual involved in review of your submission has agreed to reveal their identity: Roger J Davis (Reviewer #3).

While the reviewers found that the paper provides convincing evidence for the role of EPSTl1 in KSR1-dependent EMT, 2 of the reviewers raised valid criticisms concerning the lack of experiments to exclude an effect on degradation of EPSTl1, an important technical problem with the polysome versus monosome assignment (which could affect the interpretation of the data), and overstatements of the data. Following discussions among the referees, it was agreed that an in vivo model of colon cancer to reproduce the in vitro phenotype (as requested by reviewer #2) is not necessary for this paper. We include the full reviews of the referees because they contain in addition to the necessary changes, some very constructive suggestions for improvement of the paper.

*Reviewer #1 (Recommendations for the authors):*

1. The authors should rule out that KSR KD or KO does not impact EPSTI1 protein degradation.

2. Lines 166-167 – the authors state that KSR1-dependent translation of EPSTI1 is required for colon tumor cell transformation. First, tumor cells do not transform. They are already transformed. Second, no experiments up to this point show that EPSTI1 is required for transformation. To demonstrate this, they would need to express EPSTI1 in a normal or immortalized cell and demonstrate its ability to induce anchorage independent growth, or soft agar growth, or the ability to form tumors in mice. The experiments in figure 3 demonstrate that EPSTI1 is increased in colon cancer cell lines compared to "normal" cells and that KD of EPSTI1 in colon cancer cells leads to decreased viability and colony formation. Together these experiments prove that EPSTI1 is necessary for established cancer cell growth, but not transformation as stated.

3. Figure 4 – With regards to the impact of EPSTI1 KD or KO on cell migration, the authors should consider measuring the impact of EPSTI1 KD or KO on cell proliferation, since this could have a significant impact on the interpretation of the wound healing assay and Matrigel invasion assay. If there is no difference in proliferation then the authors' assertion that EPSTI1 is necessary for the migratory potential of the CRC cell lines hold true. However, if KD or KO cells are proliferating less (as supported by their previous work), then another reason the cells may appear to be closing the gap slower, or invading through Matrigel slower, is a decrease in cell cycle progression. If this is the case, the authors can still make the argument that EPSTI1 KD or KO cells move slower if they conduct realtime imaging of individual cells and track their movement (can be shown as a spider plot).

4. Lines 196-198 – the authors state that "the switch of E-cadherin to N-cadherin expression promotes the progression of migratory and invasive behavior orchestrated by KSR1-EPSTI1 signaling in CRC cells." This is an overstatement based on the evidence provided in Figure 5. The authors demonstrate that a switch in E-cad and N-cad are associated with migratory potential, but do not demonstrate that the switch actually promotes the progression. In order to do this, they would need to KD E-cad and/or overexpress N-cad and demonstrate that this can rescue the decrease in cell migration see in Figure 4.

5. Figure 6 – Does the enforced expression of EPSTI1 increase proliferation in these cells? If so, would consider comments in point #2.

*Reviewer #2 (Recommendations for the authors):*

1. The data shown to support the conclusion that the regulation of EPSTI1 by KSR1 is under translational control requires clarification. Authors state in the methods that RNAseq was performed on: "total RNA and RNA pooled from the polysome fraction (fractions 6-9)". According to the absorbance profile shown in figure 1A, fractions 6-9 would not correspond to efficiently translated mRNAs (i.e. bound by 3 or more ribosomes), it looks like fraction 6 is the monosome. Can the authors please clarify, as this is an important technical concern which is of course relevant to the RNA-seq analysis that the authors performed to identify EPSTI1 (and potentially other targets listed) as being under translational control. Related comment regarding Figure 1F, the comparison being made is not entirely clear, the difference between control cells and the KSR1 knock-downs seems to only be obvious in one fraction, #7, which from the absorbance profile seems to be a "light fraction".

2. Data with SCH772984 supports the importance of ERK1/2 activation in the regulation of EPSTI1 protein expression. However, the data that ERK1/2 inhibition alters the translation of EPSTI1 mRNA is not robust. The absorbance profiles related to the polysome fractionation done on HCT116 cells treated or not with SCH772984 are unusual, and thus the data shown in Figure 2E are in question. Related to EPSTI1 expression being under translational control, can its translation be repressed by an inhibitor of mTOR?

3. The authors have shown data that the KSR1-dependent regulation of EPSTI1 is essential for EMT, including effects on invasion, anchorage-independent cell viability, and the cadherin switch in colon cancer cell lines depleted of KSR1 (and phenotypes are rescued with add-backs of KSR1 or EPSTI1). Missing from the manuscript is an in vivo model to show whether the observed in vitro phenotypes (i.e. in colon cancer cells depleted for KSR1 or EPSTI1) have relevance in an in vivo model. HCT116 for example have been used in orthotopic mouse models, with HCT116-derived tumors showing evidence of local invasion and metastasis to distant organs (lung, liver). Using the cell lines already developed by the authors, in vivo characterization of the impact of loss of KSR1 or EPSTI1 can thus be performed.

4. Related to point #3, what is the expression of KSR1 in HCEC non-transformed cells compared to HCT116/HCT15? HCECs can be induced to undergo EMT (with loss of epithelial markers and gains in mesenchymal markers, including fibronectin, vimentin, and N-cadherin). Thus, does KSR1 overexpression, or EPSTI1 overexpression in HCECs promote a full or partial EMT, and increase their invasive/metastatic behaviour in vivo? The experiments in #3 and #4 are essential to understand how at what point during the progression of colorectal cancer KSR1 expression is dysregulated, and may help design future therapeutic targeting windows.

*Reviewer #3 (Recommendations for the authors):*

This is a very strong paper that provides a thorough and definitive analysis of the role of EPSTI1 in KSR1-dependent EMT. The major question related to this study is the molecular mechanism of EPSTI1 function – which should be addressed in subsequent studies and reported in a new paper.

There is one point that the authors could address. In the Discussion section of the manuscript, the authors speculate that it is possible that KSR1 changes EPSTI1 mRNA splicing. Since the authors have RNA-seq data that could address this point, the authors should refer to thus analysis if a conclusion can be drawn. If no conclusions can be drawn, no changes to the manuscript are required.

[Editors' note: further revisions were suggested prior to acceptance, as described below.]

Thank you for resubmitting your work entitled "KSR1-and ERK-dependent Translational Regulation of the Epithelial-to-Mesenchymal Transition" for further consideration by *eLife*. Your revised article has been reviewed by two peer reviewers, and the evaluation has been overseen by Jonathan Cooper as the Senior Editor and a Reviewing Editor.

The manuscript has been improved but there are some remaining issues that need to be addressed, as outlined below:

There are 2 issues which need to be addressed.

1. In Figure 6 (supplement 1), you make the point that KD of EPSTI1 does not impact cell proliferation (in D and E). This seems to be true for SW480 cells, but it is less clear for HCT116 cells. You should provide statistics for these results.

2. You make the case that enforced expression of EPSTI1 (in the context of KSR1 KO) does not increase cell proliferation. This seems to be true for HCT116 cells, but is a lot less clear for SW480. You should provide statistics for these results.

---

## [Author Response]

Reviewer #1 (Recommendations for the authors):1. The authors should rule out that KSR KD or KO does not impact EPSTI1 protein degradation.

To determine if KSR1 promotes EPSTI1 degradation, we first assessed EPSTI1 turnover in HCT116 cells following treatment with a protein synthesis inhibitor, cycloheximide (CHX) and observed that EPSTI1 has a 6-hour half-life **(**Figure 1—figure supplement 2D). We analyzed EPSTI1 turnover using a combination of proteasome inhibitor, MG132 and CHX in control and CRISPR-targeted KSR1 HCT116 cells (Figure 1—figure supplement 2E). EPSTI1 turnover was not sensitive to MG132 treatment in HCT116 cells lacking KSR1 expression. Therefore, in HCT116 cells, KSR1 does not mediate ubiquitin proteosome system (UPS)-mediated degradation of EPSTI1. These data support our conclusion that EPSTI1 translation is induced by KSR1. Our findings are presented in the results (lines 134-142), Figure 1—figure supplement 2D-E and the Discussion section.

2. Lines 166-167 – the authors state that KSR1-dependent translation of EPSTI1 is required for colon tumor cell transformation. First, tumor cells do not transform. They are already transformed. Second, no experiments up to this point show that EPSTI1 is required for transformation. To demonstrate this, they would need to express EPSTI1 in a normal or immortalized cell and demonstrate its ability to induce anchorage independent growth, or soft agar growth, or the ability to form tumors in mice. The experiments in figure 3 demonstrate that EPSTI1 is increased in colon cancer cell lines compared to "normal" cells and that KD of EPSTI1 in colon cancer cells leads to decreased viability and colony formation. Together these experiments prove that EPSTI1 is necessary for established cancer cell growth, but not transformation as stated.

We apologize for the incorrect use of the word ‘transformation’ in the sentence. We rectified the statement and changed the sentence in lines 184-185 to ‘These observations show that KSR1-dependent translation of ESPTI1 is required for anchorage-independent growth of colon tumor cell lines.’

3. Figure 4 – With regards to the impact of EPSTI1 KD or KO on cell migration, the authors should consider measuring the impact of EPSTI1 KD or KO on cell proliferation, since this could have a significant impact on the interpretation of the wound healing assay and Matrigel invasion assay. If there is no difference in proliferation then the authors' assertion that EPSTI1 is necessary for the migratory potential of the CRC cell lines hold true. However, if KD or KO cells are proliferating less (as supported by their previous work), then another reason the cells may appear to be closing the gap slower, or invading through Matrigel slower, is a decrease in cell cycle progression. If this is the case, the authors can still make the argument that EPSTI1 KD or KO cells move slower if they conduct realtime imaging of individual cells and track their movement (can be shown as a spider plot).

We agree that quantification of cell migration and invasion can be impacted significantly by cell proliferation. We confirmed that the effects observed are due solely to cell migration using two different techniques:

1. The cell migration experiments were performed using IncuCyte scratch wound analyzer. IncuCyte software is designed specifically to experimentally quantify the effects of cell migration separate from changes that occur as result of cell proliferation. Relative wound density is calculated as the percentage of spatial cell density inside the wound relative to the spatial density outside of the wound area at a given time point. The metric is designed to be zero at t=0 and 100% when cell density inside the wound is the same as the cell density outside the initial wound. The clarification on this method is included in Results section lines 190- 194 and the methods section.

2. To determine the effect of EPSTI1 on cell proliferation, we analyzed the cell growth kinetics in HCT116 and SW480 cells (Figure 6—figure supplement 1). Over a period 3 days, EPSTI1 knockdown had no effect on proliferation in comparison to control HCT116 and SW480 cells. Although, KSR1-CRISPR knockout cells grew slower that the control HCT116 and SW480 cells, EPSTI1-expression did not rescue the proliferation in these cells (Figure 6—figure supplement 1). Since cell migration was measured over 48 hours and invasion was measured for 24 hours, any effect of cell proliferation should not impact our data. Our observations are included in result section lines 238-248 and Figure 6—figure supplement 1.

4. Lines 196-198 – the authors state that "the switch of E-cadherin to N-cadherin expression promotes the progression of migratory and invasive behavior orchestrated by KSR1-EPSTI1 signaling in CRC cells." This is an overstatement based on the evidence provided in Figure 5. The authors demonstrate that a switch in E-cad and N-cad are associated with migratory potential, but do not demonstrate that the switch actually promotes the progression. In order to do this, they would need to KD E-cad and/or overexpress N-cad and demonstrate that this can rescue the decrease in cell migration see in Figure 4.

We subjected control HCT116 cells and HCT116 cells overexpressing N-cadherin to Transwell invasion assays through Matrigel following EPSTI1 knockdown. EPSTI1 knockdown suppressed cell invasion. However, the expression of N-cadherin in cells lacking EPSTI1 was sufficient to restore invasiveness to HCT116 cells **(**Figure 5—figure supplement 1C-D). Our observation is consistent with previous observations that upregulation of N-cadherin enhances motility in multiple cancer cell lines (Hulit et al., 2007; Mrozik et al., 2018; Nieman et al., 1999). These results indicate that the switch of E-cadherin to N-cadherin expression promotes the progression of migratory and invasive behavior orchestrated by EPSTI1 signaling in CRC cells. The results are included in lines 218-225 and Figure 5—figure supplement 1C-D.

5. Figure 6 – Does the enforced expression of EPSTI1 increase proliferation in these cells? If so, would consider comments in point #2.

We analyzed the cell growth kinetics in control and EPSTI1 knockdown in HCT116 and SW480 cells (Figure 6—figure supplement 1). In comparison to the control HCT116 and SW480 cells, EPSTI1 knockdown had no effect on cell proliferation over 5 days. Our observations are included in result section lines 238-245 and Figure 6—figure supplement 1*.*

Reviewer #2 (Recommendations for the authors):1. The data shown to support the conclusion that the regulation of EPSTI1 by KSR1 is under translational control requires clarification. Authors state in the methods that RNAseq was performed on: "total RNA and RNA pooled from the polysome fraction (fractions 6-9)". According to the absorbance profile shown in figure 1A, fractions 6-9 would not correspond to efficiently translated mRNAs (i.e. bound by 3 or more ribosomes), it looks like fraction 6 is the monosome. Can the authors please clarify, as this is an important technical concern which is of course relevant to the RNA-seq analysis that the authors performed to identify EPSTI1 (and potentially other targets listed) as being under translational control. Related comment regarding Figure 1F, the comparison being made is not entirely clear, the difference between control cells and the KSR1 knock-downs seems to only be obvious in one fraction, #7, which from the absorbance profile seems to be a "light fraction".

This is an important and attentive catch that we appreciate greatly. In our original polysome profile graphs we did not account for the dead volume between measurement of absorbance and the collection of each of the fraction which causes a delay of approximately 30 seconds. Therefore, the labeling of the graphs did not accurately correspond to the correct fraction number. We have rectified the labeling and shifted the graph to the correct fraction number. The corrected graphs are included in Figure 1A and Figure 1—figure supplement 1 includes all the individual experiments used for RNA seq analysis and the RT-qPCR for quantifying *EPSTI1* and *HPRT* mRNA.

For whatever reason, *EPSTI1* mRNA is predominantly located in a single polysome fraction in HCT15 cells. This was true for each of three experiments in which were isolated from HCT15 cells (Figure 1—figure supplement 1). We provide an alternate representation of that combines Figures 1F and S2C as bar graphs comparing *EPSTI1* and *HPRT* mRNA in low molecular weight fractions and high molecular weight fraction (Author response image 1). If the reviewers believe that this graph provides better clarity, we are happy to substitute it for current Figure 1F, though our preference to show measurements within the individual fractions.

**Author response image 1. sa2fig1:** RT-qPCR analysis of *EPSTI1* and *HPRT1* mRNA levels from LMW (fractions 3-5) and HMW (fractions 6-8) of the control and KSR1 knockdown HCT116 and HCT15 cells (n=3; **, P<0.05; ***, P<0.001; ns, non-significant).

2. Data with SCH772984 supports the importance of ERK1/2 activation in the regulation of EPSTI1 protein expression. However, the data that ERK1/2 inhibition alters the translation of EPSTI1 mRNA is not robust. The absorbance profiles related to the polysome fractionation done on HCT116 cells treated or not with SCH772984 are unusual, and thus the data shown in Figure 2E are in question. Related to EPSTI1 expression being under translational control, can its translation be repressed by an inhibitor of mTOR?

We magnified the polysome-bound fractions in the absorbance profile of HCT116 cells with or without ERK inhibitor treatment (Figure 2E) to reveal the polysome-bound fraction more clearly. The translation status of mRNAs varies temporally and between cell types from the same tissue, which can have a pronounced effect on the absorbance profiles (King and Gerber, 2016; Seimetz et al., 2019).

We tested effect of mTOR inhibition on EPSTI1 expression. mTOR inhibitor, AZD8055 inhibits mTOR with 1000-fold greater specificity than other lipid kinases (e.g., PI3 kinase) (Chresta et al., 2010). Though AZD8055 robustly inhibited phosphorylation of mTOR substrate p70 S6 kinase, its ability to decrease EPSTI1 expression in HCT116 cells was weak relative to treatment with the ERK inhibitor (Figure 2C). These observations suggest the ERK affects EPSTI1 expression via mechanisms distinct from mTOR. The results are included in lines 153-158 and Figure 2C.

3. The authors have shown data that the KSR1-dependent regulation of EPSTI1 is essential for EMT, including effects on invasion, anchorage-independent cell viability, and the cadherin switch in colon cancer cell lines depleted of KSR1 (and phenotypes are rescued with add-backs of KSR1 or EPSTI1). Missing from the manuscript is an in vivo model to show whether the observed in vitro phenotypes (i.e. in colon cancer cells depleted for KSR1 or EPSTI1) have relevance in an in vivo model. HCT116 for example have been used in orthotopic mouse models, with HCT116-derived tumors showing evidence of local invasion and metastasis to distant organs (lung, liver). Using the cell lines already developed by the authors, in vivo characterization of the impact of loss of KSR1 or EPSTI1 can thus be performed.

Our future directions are aimed toward extending the in vitro findings and to understand the extent to which EPSTI1 promotes metastatic behavior in colon tumors and contributes to EMT in vivo.

4. Related to point #3, what is the expression of KSR1 in HCEC non-transformed cells compared to HCT116/HCT15? HCECs can be induced to undergo EMT (with loss of epithelial markers and gains in mesenchymal markers, including fibronectin, vimentin, and N-cadherin). Thus, does KSR1 overexpression, or EPSTI1 overexpression in HCECs promote a full or partial EMT, and increase their invasive/metastatic behaviour in vivo? The experiments in #3 and #4 are essential to understand how at what point during the progression of colorectal cancer KSR1 expression is dysregulated, and may help design future therapeutic targeting windows.

Our previous study shows that KSR1 expression is upregulated in colon cancer cell lines when compared to the non-transformed human colon epithelial cells (HCECs) (Fisher et al., 2015). We added this statement in the Results section on lines 173-175*.*

We tested the effect of ectopic EPSTI1 expression on invasion in non-transformed human colon epithelial cells (HCECs). We stably expressed MSCV-IRES-GFP or MSCV-IRES-EPSTI1-GFP in HCECs and subjected the cells to Transwell invasion assay through Matrigel **(**Figure 7B, top), and observed that EPSTI1 alone was sufficient to dramatically induce the expression of N-cadherin and double the invasive activity of HCECs (Figure 7B). These data indicate that EPSTI1 mediates N-cadherin expression to promote invasive behavior in non-transformed colon epithelial cells. The observations are included in Figure 7B and result section lines 257-263.

Reviewer #3 (Recommendations for the authors):This is a very strong paper that provides a thorough and definitive analysis of the role of EPSTI1 in KSR1-dependent EMT. The major question related to this study is the molecular mechanism of EPSTI1 function – which should be addressed in subsequent studies and reported in a new paper.There is one point that the authors could address. In the Discussion section of the manuscript, the authors speculate that it is possible that KSR1 changes EPSTI1 mRNA splicing. Since the authors have RNA-seq data that could address this point, the authors should refer to thus analysis if a conclusion can be drawn. If no conclusions can be drawn, no changes to the manuscript are required.

We are working toward determining how EPSTI1 alters N-cadherin or EMT-transcription factor expression to affect the EMT-like phenotype including evaluation of, a potential role for NF-κB signaling (Li et al., 2014), and an evaluation of EPSTI1 contribution cell migration, invasion and metastasis in vivo.

Our preliminary splicing analysis from the polysome profiling RNA seq data suggest that KSR1 regulates alternative splicing events, particularly exon skipping. However, the lack of sequencing depth from the RNA seq analysis does not allow us to draw detailed conclusions regarding its role in EMT-like behavior in general or EPSTI1 translation in particular. Our future directions are aimed toward determining if, and how, KSR1 regulates EPSTI1 alternative splicing and whether a preferential splice for of EPSTI1 is translated preferentially in CRC to promote the EMT-like phenotype.

References:

Chresta, C. M., Davies, B. R., Hickson, I., Harding, T., Cosulich, S., Critchlow, S. E., Vincent, J. P., Ellston, R., Jones, D., Sini, P., et al. (2010). AZD8055 is a potent, selective, and orally bioavailable ATP-competitive mammalian target of rapamycin kinase inhibitor with in vitro and in vivo antitumor activity. Cancer Research 70, 288-298.

Fisher, K. W., Das, B., Kim, H. S., Clymer, B. K., Gehring, D., Smith, D. R., Costanzo-Garvey, D. L., Fernandez, M. R., Brattain, M. G., Kelly, D. L., et al. (2015). AMPK Promotes Aberrant PGC1β Expression To Support Human Colon Tumor Cell Survival. Molecular and Cellular Biology 35, 3866-3879.

Hulit, J., Suyama, K., Chung, S., Keren, R., Agiostratidou, G., Shan, W., Dong, X., Williams, T. M., Lisanti, M. P., Knudsen, K., and Hazan, R. B. (2007). N-cadherin signaling potentiates mammary tumor metastasis via enhanced extracellular signal-regulated kinase activation. Cancer Res 67, 3106-3116.

King, H. A., and Gerber, A. P. (2016). Translatome profiling: methods for genome-scale analysis of mRNA translation. Brief Funct Genomics 15, 22-31.

Li, T., Lu, H., Shen, C., Lahiri, S. K., Wason, M. S., Mukherjee, D., Yu, L., and Zhao, J. (2014). Identification of epithelial stromal interaction 1 as a novel effector downstream of Krüppel-like factor 8 in breast cancer invasion and metastasis. Oncogene 33, 4746-4755.

Mrozik, K. M., Blaschuk, O. W., Cheong, C. M., Zannettino, A. C. W., and Vandyke, K. (2018). N-cadherin in cancer metastasis, its emerging role in haematological malignancies and potential as a therapeutic target in cancer. BMC Cancer 18, 939.

Nieman, M. T., Prudoff, R. S., Johnson, K. R., and Wheelock, M. J. (1999). N-cadherin promotes motility in human breast cancer cells regardless of their E-cadherin expression. J Cell Biol 147, 631-644.

Seimetz, J., Arif, W., Bangru, S., Hernaez, M., and Kalsotra, A. (2019). Cell-type specific polysome profiling from mammalian tissues. Methods 155, 131-139.

[Editors' note: further revisions were suggested prior to acceptance, as described below.]

The manuscript has been improved but there are some remaining issues that need to be addressed, as outlined below:There are 2 issues which need to be addressed.1. In Figure 6 (supplement 1), you make the point that KD of EPSTI1 does not impact cell proliferation (in D and E). This seems to be true for SW480 cells, but it is less clear for HCT116 cells. You should provide statistics for these results.

The statistical significance, or lack thereof, for each condition in comparison to EPSTI1 KD in HCT116 and SW480 cells is now added to Figure 6—figure supplement 1. Our observations are described on lines 238-248.

Over the first 3 days, EPSTI1 knockdown had no significant effect on cell proliferation compared to control HCT116 and SW480 cells. After 7 days (9-10 doublings), EPSTI1 knockdown decreased HCT116 cell proliferation by only 16.4 %. These same initial conditions were used to measure migration over 72 hours and invasion over 24 hours. Furthermore, the IncuCyte software corrects for the influence of proliferation on migration rate. These data show that cell proliferation would not impact quantification of motility or invasion.

2. You make the case that enforced expression of EPSTI1 (in the context of KSR1 KO) does not increase cell proliferation. This seems to be true for HCT116 cells, but is a lot less clear for SW480. You should provide statistics for these results.

Thank you for identifying our mischaracterization of enforced EPSTI1 expression on SW480 cells. The significance, for each condition in comparison to KSR1 KO in HCT116 and SW480 cells is now added to Figure 6—figure supplement 1. These observations are described on lines 238-248.

EPSTI1 expression in KSR1 knockout cells has no significant effect on cell proliferation for 24 hours in HCT116 and SW480 cells (Figure 6—figure supplement 1A-B). However, EPSTI1 expression increased the number of invading cells by over 50% in HCT116 and over 70% in SW480 cells (Figure 6D) in that period. Although, EPSTI1 expression has a significant effect on cell proliferation over 7 days compared to KSR1 knockout cells in HCT116 and SW480 cells, EPSTI1 expression was enough to rescue the migratory potential by over 60% in KSR1 depleted HCT116 and SW480 cells within 24 hours (Figure 6C). As indicated above and in the manuscript, IncuCyte cell migration software corrects for cell proliferation. Therefore, the cell proliferation should not impact the quantification of motility.